# Is domestic agricultural production sufficient to meet national food nutrient needs in Brazil?

João Pompeu[1]◉*, Camille L. Nolasco[1]◉, Paul West[2]‡, Pete Smith[3]‡, Jacqueline Gerage[1]‡, Jean Ometto[1]‡

**1** Earth System Science Centre, National Institute for Space Research, São José dos Campos, São Paulo, Brazil, **2** Institute on the Environment, University of Minnesota, Saint Paul, Minnesota, United States of America, **3** Institute of Biological and Environmental Sciences, University of Aberdeen, Aberdeen, United Kingdom

◉ These authors contributed equally to this work.
‡ These authors also contributed equally to this work.
* joao.pompeu@inpe.br

**Data Availability Statement:** All relevant data are uploaded to the PANGAEA database and publicly accessible via the following URL: https://doi.pangaea.de/10.1594/PANGAEA.911574.

## Abstract

Reducing the impacts of agriculture on the environment is one of the greatest challenges of this century. In Brazil, it is often argued that more land use change is needed to achieve food security. However, analyses seeking to understand the dynamics between agricultural production for exports and food intended for the Brazilian population have not approached the question if national agriculture is sufficient to provide Brazilians with the necessary nutrients, according to nutritional recommendations. In this sense, we sought to combine supply and dietary requirements for food (calories and nutrients) to assess trends in nutrient production and how future population projections and possible changes in diets would affect land necessity for nutritional security. We use sub-national data on agricultural production, population, Food Balance Sheets from FAO, and a compilation of nutritional information on the Brazilian agricultural production. Our results show that, in the last three decades, Brazil produced enough food calories to feed on average 115% of its population. We found that the agricultural land in 2017, without any expansion, is sufficient to feed, at least, 105% of projected population in 2060, considering the same productivity and dietary patterns. In a vegan diet scenario, less than 10% of the land dedicated to agricultural production in the past 30 years would be required. Despite limitations on supplying certain micro-nutrients, a vegan diet would require even less land in the future. We conclude that Brazilian agriculture could deliver enough food to meet Brazilians' nutritional needs without further land expansion. Food production is compatible with environmental conservation in Brazil, especially if meat consumption is reduced.

**Funding:** J. P. received his doctorate scholarship from Coordenação de Aperfeiçoamento de Pessoal de Nível Superior (CAPES). J. P., J. G. and J. O. were funded by the Fundação de Amparo à Pesquisa do Estado de São Paulo (FAPESP), in the scope of the Belmont Forum FACCE-JPI 2013 (process 2014/50627-2). The funders had no role in study design, data collection and analysis, decision to publish, or preparation of the manuscript.

**Competing interests:** The authors have declared that no competing interests exist.

# Introduction

One of the major environmental challenges in this century is to feed a growing population without increasing the pressure on land resources [1, 2]. Expansion and intensification of agriculture since the 1960s was successful in augmenting food production, but it also caused numerous adverse environmental impacts [3]. Important impacts from agriculture, though not limited to these, are habitat loss and fragmentation, with almost 40% of the ice-free land surface used for agriculture [4], growth in nitrogen fertilizer and water use of 800% and 100%, respectively [5] and emission of 23% of the total anthropogenic Greenhouse Gases (GHG). Especially, GHG emissions contribute to the current increase in global average air temperature between the pre-industrial period to present day [6], which might lead to a negative feedback for agricultural production in several regions [6].

However, more than 820 million people globally are still hungry and undernourished, underscoring the immense challenge of achieving the Zero Hunger target by 2030 [7]. In this sense, agricultural trade is important for achieving global food security [8]. Nearly a quarter of all food that is consumed in the world is traded [9], with 20 major exporting and 33 major importing countries, that account for 70% of global trade [10]. Among the largest exporters, Brazil contributes nearly 10% of the global land cropped for the international trade in agricultural commodities [10]. This makes it the third largest exporter in the world, after the European Union and the United States and highlights its role in the global food system [11].

Brazil holds the greatest share of South America's biodiversity [12]. It has the largest tropical forest area and carbon stock in the world [13], thereby contributing to regional, continental and global regulation of several ecosystem services [14, 15]. Around a quarter of the country's territory is currently under agricultural production [16] and regional climate change, driven by land use change, can already be observed in the most recent agricultural frontiers, e.g. the Amazon [17, 18]. Thus, additional large-scale land use changes are likely to cause negative environmental impacts with global implications for climate change via GHG emissions. Soil erosion, nutrient loss, water depletion and biodiversity loss [19, 20] are important regional impacts of land use changes. Agricultural yields are also affected by these impacts, increasing the demand for more land for production, leading to the positive feedback [21]. As already demonstrated in Brazil, extensive land clearing for agriculture in the Cerrado and eastern Amazon alters regional weather, reducing maize yields by 6% to 8%, undermining the intended use of the cleared land [22].

Land use expansion for growing commodity crops is an important driver in promoting deforestation and GHG emissions in Brazil [23, 24], although law enforcement and agricultural intensification decoupled this trend in the mid-2000s [25, 26]. Some of the central policies for environmental protection and reduction of deforestation, in Brazil, were the Action Plan for Prevention and Control of Deforestation in the Legal Amazon (PPCDAm) launched in 2004 [27] and later implemented in the Cerrado region (the PPCerrado), a key hotspot for biodiversity and agricultural production [19]. The Soy Moratorium, a commitment of the private sector to not purchase soybeans from deforested areas also had positive effects on illegal deforestation in the Amazon [28].

Despite the introduction of legislation and governmental actions to reduce deforestation in Brazil from 2004 to 2012, significant changes in the environmental legislation were made under pressure from the agribusiness sector [29]. At the same time, the government reduced its public budget for environmental actions and encouraged agribusiness expansion with financial support. This resulted in increasing deforestation trends since 2012, reaching in 2020 its highest peak since 2008 [30–32]. Political efforts to reduce environmental protection in Brazil were often based on the argument that more land is needed to produce food for the

Brazilian population [29–34]. This argument was central to the debate that changed the environmental legislation in 2012 and is still argued in some proposals, both from the national congress and government, for promoting the agricultural expansion [29]. However, the proportion of land needed in the territory for meeting the nutritional food demand of the Brazilian population, currently around 210 million people, is unknown. Increasing the productivity of pasturelands could negate the need for any further land to be deforested land for food production [33–35]. Additionally, more food can be delivered on the same amount of land if dietary preferences are changed [36–38]. Enriching diets with plant-based foods and with fewer animal sources is argued to be beneficial for both human health and for the ecosystems [39], reducing the environmental pressures of food production on the environment. Shifting diets towards those which contain lower quantities of livestock products could reduce the amount of land needed to produce food, in terms of calories, macro- and micro-nutrients for human consumption. Taken together, increasing yields through sustainable intensification and changing diets could make a significant contribution to deliver food security while reducing the pressure on the environment due to land use expansion.

In this context, we assess whether Brazilian agriculture could provide sufficient nutrients for the Brazilian population through its domestic production with the current land use for agriculture. Focusing more specifically on the production side of food security (also referred to as food availability in food security literature), we combined supply and dietary requirements for calories, macro- and micronutrients based on three decades of data on agricultural production and trade, land use and nutritional requirements of the population. We then assessed how future population projections and changes in diets would affect land demand for food security. Our study provides evidence that no additional land use changes are needed to meet nutritional supply in Brazil. Shifting diets towards less meat consumption would save a considerable amount of land, since pasture for livestock production is largely the more extensive anthropogenic land use in the country, while most of the food nutrients and calories are produced from vegetable crops.

## Materials and methods

Our study was primarily based on the methods of [38], which allocated worldwide agricultural data into national domestic allocations for food and non-food uses and exports. Adapting this approach to a temporal analysis for Brazil, we used six main sources of data, at national and sub-national (municipality) level:

- crop, livestock, eggs and dairy production, from the National Institute for Geography and Statistics (IBGE) [40, 41]. All of the 62 crops and 5 livestock products available since 1988 were considered;

- the Food Balance Sheets (FBS) from the Food and Agriculture Organization (FAO);

- land use and cover maps from MapBiomas [42];

- the nutritional content of crops, livestock and dairy from the Brazilian Table of Food Composition (TACO) [43] whenever they were available, otherwise we used The United States Department of Agriculture (USDA)'s National Food and Nutrient Analysis Program (NFNAP) values [44];

- Recommended Dietary Allowance (RDA) and Adequate Intake (AI) of nutrients, from the United States Institute of Medicine [45] and;

- IBGE's national population projection [46–48].

The national surveys PAM and PPM (the Portuguese acronyms for Municipal Agriculture Production and Municipal Livestock Production) accounts for area, production, and productivity of 62 crops, cattle, pork, poultry and animal-derived products, like eggs and milk, in all of the 5,572 Brazilian municipalities, from 1988 to 2017. This information is collected by IBGE agents and the accuracy is checked to ensure inter-annual consistency. These data are the source of the Brazilian data to FAO, however the values reported in the PAM are not aggregated, as in the FBS, into "Export", "Feed", "Fibre", "Seed", "Losses", "Processed", "Other Uses" and "Food". In addition, the FBS also accounts for the imports of agricultural products. Thus, we used sub-national agricultural production from PAM and PPM, as well as the national level FBS to estimate the proportion of agricultural production that is exported or that stays in Brazil as food and feed for domestic dietary requirements, considering that the fractions of food, feed and exports are proportional and evenly distributed to the municipality they are produced.

Additionally, we obtained crop and pastureland areas for every year from 1988 to 2017 from MapBiomas's land use and land cover maps (collection 3.0). MapBiomas is a partnership of private companies, universities and NGOs that use all available series of remotely sensed images from Landsat satellites to produce land use and land cover maps for the Brazilian territory, in the period of 1985 to 2017, with 30 metres of spatial resolution [42]. Based on the municipal data from PAM, the class "Crop or Pasture", an error class in MapBiomas that cannot be distinguished due to spectral confusion, was allocated to "Pasturelands" or "Croplands" classes, adjusting the cropland maps to the official Brazilian cropland area. This allowed us a yearly estimate of the pasture area at the national level.

We used three of IBGE's population projections to ensure the best yearly population data in the study period. The first, from 1980 to 2050 [46], was used to account for the population from 1988 to 1999. Then, we used the projection from 2000 to 2060 [47] for the period of 2000 to 2009 and, finally, the projection from 2010 to 2060 [48] for the period from 2010 to 2017 and then to a scenario analysis until 2060.

## Disaggregating agricultural production

In this step, we first calculated the amount of annual calorie, macro- and micro-nutrients production for each category: crops, livestock, milk and eggs, based on TACO/NFNAP values. Then, based on [38], we disaggregated the calories to "Export", "Feed", "Fibre", "Seed", "Losses", "Processed", "Other Uses" and "Food", according to the proportion of these categories reported by the FBS yearly.

As in [38], some assumptions had to be made to allocate some products according to the domestic use (e.g. "Other Uses" of Cottonseed that were considered as "Fibres", while the "Processed" proportions of Groundnut, Rice, Barley, Coconuts, Maize and Grapes were allocated proportionally as food or feed). A list of assumptions for the allocation of the products is provided in the S1 Table. Crops omitted from this table were allocated exactly as reported in the FBS for each specific crop.

FAO's Technical Conversion Factors for Agricultural Commodities [49] was used to convert livestock weight to edible proportions of beef, pig and poultry meat, which were 71%, 73% and 78%, respectively. We assumed that both beef and milk cattle were produced in grazing and not confinement systems, which accounts for only a small fraction (2.4%) of cattle production in Brazil [50]. Assuming that all of the lactating cows are milked and following the proportion of non-lactating and lactating (∼35%) cows in the national dairy cattle [51], we calculated land use for milk as three times the average of cattle/ha multiplied by the amount of

milked cows from PPM. Therefore, both non-lactating and lactating cattle is considered in the pastureland area for milk production as they compose the national dairy farming system.

Since crop and livestock production were available in Brazil at sub-national level until 2017 and the FBS only until 2013, by the time of this analysis, we modelled the proportions of food, feed, exports etc. for the period of 2014 to 2017 using linear regression with IBGE's production as the predictor. To minimize the effects of the trend in the time series, and to ensure that the best regression parameters were used, for each of the agricultural products we ran 24 different regressions comprising different periods, i.e. 1988–2013, 1989–2013, 1990–2013. . . 2011– 2013, and the model with the higher adjusted $R^2$ was selected to predict food, feed, exports and processed products. The adjusted $R^2$ was used for model selection because it considers the sample size, thus it was a suitable parameter to compare the models.

According to the United States Institute of Medicine, there is no RDA for energy because energy intakes above the Estimated Energy Requirement would be expected to result in weight gain [45]. Thus, the proportion of the population potentially fed with national food production was estimated based on a diet of 2,450 kcal/person/day (894,250 kcal/yr), which is the maximum value of Average Dietary Energy Requirements (ADER) recommended by FAO for Brazil. ADER is the individual's dietary energy requirement and it is used as a normative reference for nutrition in the population. The ADER value refers to the food deficit, that is, the amount of dietary energy needed to ensure that hunger would be eliminated if food was properly distributed. Other energy input values are found in the literature (e.g. [38] considered 2,700 kcal/caput/day, which is based on the average global calorie consumption, as described by [52], while [53] considered 2,353 kcal/caput/day which is the global ADER). Since this is a national evaluation, we used the specific requirement for Brazil. Following [54] and [55], we also considered 3 macro-nutrients (carbohydrates, fibres, and proteins), 8 vitamins (Vitamin A, Vitamin C, Thiamine, Riboflavin, Niacin, Vitamin B6, Vitamin B12 and Folate) and 8 minerals (Calcium, Copper, Iron, Magnesium, Manganese, Phosphorus, Zinc, Potassium), that we had information on nutritional content. This allows us to assess the broader aspects of food security and nutrition, beyond only calories and proteins. Caloric and nutritional production for the municipalities in the period, as well as detailed methods, are found in [56].

We used the nutrient requirements for a representative consumer, weighting each age and gender group by the population projections from IBGE [46–48], based on the measures developed by the U.S. Institute of Medicine. Following the methods from [54], whenever available, we use the Recommended Dietary Allowance (RDA) which represents the average intake that would meet the needs of 97.5% of healthy individuals in a group. If an RDA was not available for certain nutrients, we used Adequate Intake (AI), which is the level of intake assumed to be adequate for healthy individuals. The daily requirements for each nutrient in each year, weighted for the Brazilian population, is found in the S2 Table. This way, we had both nutrients and calories averaged for the structure of the Brazilian population every year in the period of analysis. Caloric and nutrient requirements were calculated based on the population projections and did not consider changes in income or individual preferences.

We calculated two forms of nutritional yields found in the literature. The first, based on [38] accounts for the number of people that could meet 100% of their energy from a hectare. In our case, we used the ADER as energy reference and call this approach as "people fed per hectare". The second, based on [55], refers to the number of people who would be able to obtain 100% of their requirements of different nutrients per hectare of land. Our study considers 19 macro- and micro-nutrients, thus we call "people nourished per hectare" the number of Brazilians able to obtain the requirements for these nutrients in one year from one hectare.

We then built different scenarios of changing diets towards less meat consumption to estimate the amount of land needed to feed the Brazilian population, based on the dietary

requirements, as discussed in [57], of the population as projected by IBGE [48], with more detailed assumptions in the S3 Table. The first scenario, the Business-as-Usual (BAU), is the current Brazilian domestic dietary requirements without any changes; the second scenario considers no beef consumption; the third scenario is a lacto-ovo vegetarian diet, where a proportion of feed to produce eggs and dairy is maintained, based on the nutritional efficiency conversion of eggs and dairy, and the rest of feed is proportionally allocated into exports and food. Finally, the fourth scenario is a vegan diet, with no animal calories, where feed is also proportionally allocated to human food and exports. On average, 49.1% of the feed is from maize (±4.8%), followed by soybeans (36.2% ±11.8%) and cassava (14.1% ±4.8%). Out of the staples in Brazil, cassava and maize are the second and third most important in terms of energy supply, after rice. We used 2017 food production as the baseline for the future projections, assuming that any additional land is used after this year. This simple assumption is adopted to test the claim that more land is needed to feed the Brazilian population.

## Results

### Caloric production

Brazilian agriculture noticeably changed from 1988 to 2017. Croplands area increased from 56.7 to 78.9 million ha (increase of 39.1%), pasturelands from 126.9 Mha to 166.6 Mha (increase of 28.5%), and the total farmed land expanded by 33.7%, from 183.6 Mha to 245.5 Mha in this period. This number does not include tree plantations for fibre and cellulose (mainly *Eucaliptus spp.*, *Pinnus spp.* and *Acacia spp.*), that expanded from 1 Mha to 6 Mha in the period and had no direct contribution in terms of food production. In this period, national calorie production increased by 203%, from 0.4x1015 to 1.3x1015 kcal, while the population grew 46.3%, from 141 million to 206 million people. Crop calories increased threefold ($4.28 \times 10^{14}$ kcal to $1.28 \times 10^{15}$ kcal), showing a pronounced pattern of crop yield intensification in the country (cropland area increased only 39% in the same period). Animal calorie production increased fourfold, from $1.9 \times 10^{13}$ kcal to $7.7 \times 10^{13}$ kcal.

Such an increase in crop production was mainly driven by export commodities with a less pronounced increase in food crops. The amount of food crop calories increased by 19%, from $1.40 \times 10^{14}$ to $1.67 \times 10^{14}$ kcal, while its proportion in total production declined from 32.8% to 13.0% (Fig 1). On the animal production side, more than 75% of the calories are for food, in the entire time series. Feed fractions are stable over time, with an average of 12.2% of the total

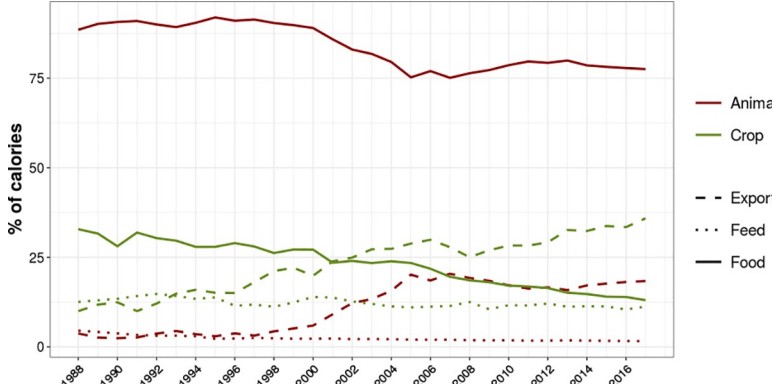

**Fig 1. Caloric production in Brazil from 1988 to 2017.** Animal (red) and crop (green) calories sum 100% each, disaggregated into calories for export (dashed lines), domestic food consumption (solid lines) and animal feed (dotted lines). Wastes and other uses are not shown.

crop calories and less than 5% of animal calories, which are derived from dairy products. On the other hand, the amount of plant calories that are exported increased more than tenfold, from $0.43 \times 10^{14}$ to $4.60 \times 10^{14}$ kcal (9.9% to 35.8% of the total crop calories) and animal calories for export increased eighteenfold, from $7.4 \times 10^{11}$ to $1.4 \times 10^{13}$ kcal (3.7% to 18.3% of the total animal calories), though in much lower absolute values than in exported crops. In 1988, calorie exportation (both plant and animal) was equivalent to 27% of the food calories. Since 2002 Food calorie exports surpass domestic consumption. Fifteen years later, in 2017, the total exported calories were equivalent to 208% of domestic food calories.

Ninety one percent (91%) of the planted area in 2017 was occupied by only 8 crops: Soybeans, Maize, Sugar Cane, Rice, Cassava, Beans, Wheat and Coffee. Soybeans alone cover 43% of the croplands and along with Maize (which commonly is planted in rotation with soybean, in a double cropping system) and sugarcane, these three crops correspond to 78% of the harvested area. In 2017, five main staple crops delivered 69% of the plant calories to the Brazilian food system: Rice (34.5%; $36.4 \times 10^{12}$ kcal), Cassava (9.7%; $10.2 \times 10^{12}$ kcal), Maize (9.0%; $9.57 \times 10^{12}$ kcal), Beans (8.6%; $9.08 \times 10^{12}$ kcal) and Wheat (6.9%; $7.32 \times 10^{12}$ kcal). On the other hand, Soybeans, Sugarcane and Coffee were responsible for, respectively: 4.5%, 2.1% and less than 0.1%. Brazil historically exports most of its coffee production ($\sim$75% in the period) and since 2013, more than 50% of the Soybeans are exported. However, within the five main staples, only maize is nowadays largely exported (>25% in 2017), while 5.4% of the wheat and 4.9% of the rice were exported in 2017. Less than 1% of the production of the other staples is exported. Maize, cassava, sugarcane (leftover parts) and soybeans accounted for $\sim$98.0% ($\pm$0.005%) of the national crop for animal feed.

## Macro- and micro-nutrient production and diet scenarios

The Brazilian agriculture could feed 5.8 people per ha of farmland in 2017 with all the calories it produces. However, as only a fraction of these calories is produced as food, it is sufficient for less than one person per ha, taking into account both the production of plant and animal food products. Considering the production of plant and animal food products in the time span between 1988 to 2017, on average 0.96 ($\pm$0.06) persons could be fed per ha per year. There has been no change in the number of people fed per ha in the last 30 years. At the same time, since 2002 there was a pronounced shift towards an intensification of the agricultural system that almost doubled the calories produced per ha of farmland. This provides evidence that the agricultural sector as a whole is shifting to market-oriented commodity production, while the food calorie increase at the same rate as the population. As discussed below, however, increasing food production does not reflect an even distribution of food for every person.

When taking the macro- and micro-nutrients into account, the overall agricultural production could nourish, on average in the period, 1.31 caput/ha ($\pm$0.19) and the fraction delivered as food could nourish 0.86 caput/ha ($\pm$0.07), increasing from 0.74 caput/ha in 1988 to 0.99 caput/ha in 2017. The difference of 10% less people nourished compared to the number of people fed per hectare reveals the prevalence of a more calorie-rich food production.

However, food production was not the major issue for food security in Brazil in the period of 1988 to 2017, as shown in Fig 2, which illustrates how much of the population could be fed with the calories produced nationally. Neither is it projected to be a problem in the future, considering the 2017 baseline of agricultural production. From 2003 onwards, national food production (BAU in Fig 2) could feed at least 120% of the population, which was also true in 1988 and 1989 (>125%). In 1990 and in 2001, national food calories could meet 105.9% and 106.8% of the population's dietary requirements, respectively.

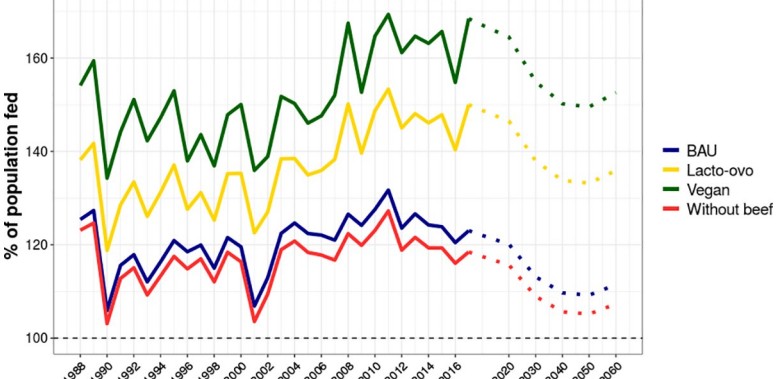

**Fig 2. Percent of the Brazilian population potentially fed with different diets in the past and projected to 2060 using the food production of 2017 as the baseline for the future scenarios and IBGE [48] population projection.** Caloric requirements is based on the maximum FAO's Average Dietary Energy Requirements (ADER) recommendation for Brazil, which is 2,450 kcal/caput/day (893,520 calories per year per person). Dashed black line is when 100% of the population is fed and the coloured dotted lines are the future projections to 2020, 2030, 2040, 2050 and 2060, using the production of 2017 as the future baseline. Complete assumptions of BAU (blue), No-beef (red), Lacto-ovo (yellow) and Vegan (green) scenarios are provided in S3 Table.

Out of the macro- and micro-nutrients, we found that vitamin A is the most limiting one in our analysis, meeting the nutritional requirements of 98% of the population in 1988, 1989, 1990 and 1993 and more than 100% in the other years. From 2003 to 2017, the availability of vitamin A increased more rapidly, to meet the nutritional needs of 106% to 131% of the population, respectively, with a peak of 137% in 2014. In turn, since 2009 the produced vitamin A was sufficient for more people than the produced calories. Therefore, all the nutrients were sufficient to nourish more people than the requirements met by the calories alone. Thus, the Brazilian agricultural sector delivers far more calories and nutrients than necessary for the domestic food dietary requirements, meaning that food security in the country is a matter of food access, and/or utilization. S4 Table shows the percent of population with energy and nutrient requirements met in the different scenarios.

We evaluated shifts in diets towards lower levels of meat consumption and calculated the proportion of the population potentially fed based on IBGE's population projection [48]. The Brazilian population is projected to grow until 2047, to >233 million people and then it is projected to reduce to around 228 million people in 2060, back to the levels of 2034. As food production also increased, even though slowly, more than 120% of the population could be fed with all the diets by 2010, while a vegan diet could feed 126 million more people. By 2050, Brazil could feed 150% of its population with a vegan diet, and 105% without any changes in the patterns of production (BAU in Fig 2). It is important to highlight that in the lacto-ovo vegetarian scenario the availability of vitamin B12 could meet nearly 100% of the population requirements in 2017, while in the vegan scenario both vitamin A and B12 production are 50% and 90% lower than the requirements for an adequate health in the same year. The other nutrients still meet the requirements of more people than those met by calories.

In Fig 3, by comparing the curves 'BAU' and 'Without beef', it is evidenced that beef accounts for a smaller fraction of the total available calories and nutrients in Brazil. Conversely, cattle use most of the land, which is reflected in the amount of land needed to sustain the BAU diet and the huge difference to a no-beef diet (Fig 3). Different diets in the future could reflect profound changes in land use, as pasturelands account for more than twice the area of croplands. Shifting diets to those containing lower quantities of animal products can potentially reduce land demand by 66.5% in the scenario without beef, or by 94.1% in the vegan scenario,

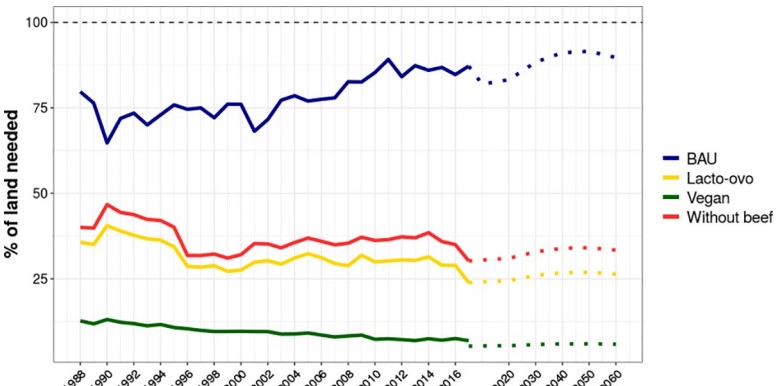

**Fig 3. Percent of land needed to feed the Brazilian population in different diet scenarios.** BAU (blue) means no change in diets (i.e. changing the population in the future but with food production levels from 2017); Dashed black line is 100% of the land in that year and the coloured dotted lines are the future projections to 2020, 2030, 2040, 2050 and 2060. Land is derived from yearly maps processed from MapBiomas (adjusted to PAM) and the projections are based on 2017 Farmland (100% from 2020 to 2060 is ∼245Mha) and the population derived from IBGE [48]. Complete assumptions of BAU (blue), No-beef (red), Lacto-ovo (yellow) and Vegan (green) scenarios are provided in S3 Table.

which would save between 165 and 231 Mha of agricultural land. Calcium and vitamin A for human consumption are mostly cattle based (up to 75% in certain years, as shown S1 Fig). Except for these nutrients, along with zinc and vitamin B12 (~30% to ~45% produced from pastures), more than 70% of the nutrients and calories are produced from food crops (S1 Fig).

In the past, Brazil could have used 10% less land to produce enough calories and nutrients for domestic supply (BAU). If food was properly distributed, even using less land for food production, would not compromise Brazilian agricultural exports. As most of the land was used for cattle ranching in low productive pastures, a lacto-ovo diet or a beef-free diet are quite similar in the amount of land that would be required for meeting the caloric requirements of the population. If cattle consumption were eliminated, less than 35% of current Brazilian agricultural land (2017 baseline of ∼245 Mha) would be required in the future, while a vegan diet would require only 14.5 Mha in 2060, or approximately 5% of the actual land used for agriculture, although the analysed crops are nutritionally poor both in vitamin A and B12.

## Assessing confidence in analysis outcomes

First, we assessed the accuracy of MapBiomas's Croplands class with the cropland area reported by PAM/IBGE, in all the 5,572 Brazilian municipalities, by means of the Root Mean Squared Error (RMSE), considering PAM as the observational values. The overall differences range from 31% in 1988 to 15% in 2017. Also, the proportion of "Crop or Pasture" class in MapBiomas maps decrease with time, possibly due to better samples and data sources in recent years, if compared to more than 30 years ago. We then adjusted the maps to PAM values by disaggregating this "Crop or Pasture", assuming PAM's area as the reference.

Total Brazilian farmlands derived from MapBiomas were 29.5% of the national territory in 2017 (251Mha), while we estimated, based on PAM and MapBiomas, that pasturelands and croplands occupied 19.5% and 9.1% of Brazilian area, respectively. According to [58], Total farmlands account for 30.2%, while croplands are 9.0% and pasturelands are 21.8%, remarkably similar to our maps for 2017. Additionally, we compared our pastureland estimates with the Digital Atlas of Brazilian Pasturelands from [59], one of the base maps for MapBiomas, and found that, on the average from 1988 to 2017, our pasture areas are 4.57% smaller than the mapped pastures. [26] found that cattle stock rates were 0.89 head/ha in 1990 and 1.36 head/ha

in 2010, while we estimated 1.09 head/ha in 1990 and 1.21 head/ha in 2010. Although our livestock data was the same from IBGE, the pastureland maps of [26] were obtained from Landstat non-forest fractions of 2000–2012 [60], which explains the much lower value they found in 1990 (0.89), when they compared the livestock from that year with a LUC map from 2000. According to [59], the year 2000 had 30Mha more pastures than 1990. So, if we divide the livestock from 1990 (147 million heads) by our 2000 pasture map (165 Mha), we find exactly the same stock rate of 0.89 head/ha. Moreover, [26] report 220 Mha of agricultural use in Brazil in 2012, 70% of which are pasturelands. This farmland estimate is 20Mha less than that in MapBiomas, but we found a similar pattern of 71.2% of pasturelands in 2012.

As our cropland and livestock estimates are derived from IBGE's PAM and PPM, the same cropland areas and trends are found for the main staples, as well as for livestock, in [33], who used the same sources.

## Discussion

Although Brazil left FAO's world hunger map in 2014 [61], food production was never a limiting factor for food security in the country since 1988, as shown in this study. The causes of food insecurity in Brazil are likely to be rooted in poverty and in the inefficient distribution and retailing of food, that constrains stable access to healthy food. This is illustrated by the fact that between 2014–16 and 2017–19, the number of moderately or severely food-insecure people in Brazil increased by 15%, reaching a fifth of the population [62]. Even while exporting more agricultural commodities and producing enough food, Brazil is in danger of again being reinstated on the world hunger map. As we focus on the production of food crops, we ignore later stages of the supply chain and the food environment, which have an important effect on the way food is consumed (e.g. through the consumption of highly processed foods) [63]. Yet, the agricultural yields and area available for food production is relevant especially when debating ecosystem services and natural resource preservation, as well as the agricultural land expansion narrative linked to food security. The development of national policies for reducing poverty towards zero hunger during the first decade of this century had significant effects on national food security, enhancing the population's access to food as one of the main strategies of a National programme [64–66]. However, with economic recession in Brazil since 2014, higher rates of food insecurity are foreseen and expected in the coming years, mainly related to greater social vulnerability of families [67]. Further, climate change may affect future agricultural production with possible impacts on food availability [8].

Nonetheless, Brazil has a unique condition to achieve food security while maintaining its export economies and protecting the country's biodiversity and natural resources [1, 2]. This is a complex issue that can be approached in multiple ways. For example, on the consumption side, shifting diets towards less meat consumption can have a sizeable impact on land demand for pastures, which occupied 19.5% of the Brazilian territory in 2017. On the production side, pasture productivity enhancement and optimised cattle stocking rates [33, 34] can reduce the demand for pastureland. There is evidence, however, that Brazilian diets are shifting in the opposite direction, towards more meat consumption [68], and to levels above those recommended by the World Cancer Research Fund [69]. The analysis of [68] reveals that income plays an important role in this pattern, not because of individual preferences to consume more meat, but due to the affordability of animal sources of protein. Strengthening food security policies that ensure the access to more diversified and fresh food could play an important role in shifting diets towards less meat consumption.

Regarding export of agricultural commodities, Brazilian international trade has higher shares of calories (9%–10% of global trade flows) than of monetary value (4%–5%) [10]. This reflects that

most of the Brazilian agricultural exports in volume are raw products with low added value, like coffee and soybeans. These products account for nearly a fifth of the country's GDP, with increasing trends. Therefore, the continued growth in export earnings seems to be the primary reason for expanded agricultural production rather than improving the people's food security. Processing these grains before export could add monetary value to the international trade. In this sense, [70] showed that agricultural expansion for export could improve the development indexes due to income and labour generation mainly in the Brazilian municipalities at the agricultural frontiers. Also, [71] claim that the expansion and improvement of this sector has generated substantial improvements in well-being throughout Brazil in several ways, such as governmental revenues for welfare programs and increasing private purchasing power. However, both works acknowledge that this is accompanied by income and land concentration, displacement of peasants and rural to urban migration with consequent losses of livelihoods when people move to marginal conditions. While the earnings from commodity exports improve local food security by increasing the economic affordability of food to more employed families, the benefits of global trade do not always necessarily overcome the social impacts. A socially-inclusive strategy and better multisectoral planning is needed for reducing exclusion in agricultural consolidated and under expansion areas.

We showed that no additional land is necessary to feed the Brazilian population in the future if current (2017) levels of production are maintained. In the BAU scenario, 220 Mha are required to produce food required by 2060, while completely shifting the diets towards a vegan diet would require only 14.5 Mha, despite limitations in vitamins A and B12 availability. Here, food fortification would play an important role for meeting the nutritional requirements of the population. Also, enhancing crop production, closing yield gaps, and reducing losses and waste are strategies to deliver higher yields and increase food availability using less land [37, 72]. The redesign of the agricultural systems is critical to limit land requirements and potential expansion over natural ecosystems, regardless of dietary change.

The Brazilian agriculture could feed 10.6 people per ha of cultivated land with all available production, based on a 2,300 kcal/caput/day diet [38]. However, it delivers only half of its calories as food, resulting in 5.2 people per ha. The estimates of [38] are based on data of circa the year 2000 (1997–2003). To compare with [38], we found quite similar values for the year 2000 (12.2 and 3.3 caput/ha, respectively). The differences in our estimates are due to the exclusion of pasturelands and the inclusion of exports in their analysis.

We used the ADER of 2,450 kcal/caput/day in our estimates, which is the maximum value recommended by the FAO to Brazil to eliminate hunger if food is properly distributed. According to [73], the average caloric consumption in Brazil is 2505.55 kcal/caput/day, a number inflated by the 9th and 10th income deciles that account for less than 1% of the population [74]. The 1st income decile (40% of the population) consumes on average 2171.37 kcal/caput/day while the 10th decile (~1% of the population) consumes on average 3215.69 kcal/caput/day. When taking the first eight income deciles into account (99% of the population), the average national caloric consumption decreased to 2442.5 kcal/caput/day. We thus consider that the ADER recommended by FAO to Brazil is also a good representation of the average national energy consumption. Additionally, other similar studies usually estimate the number of adults whose nutritional requirements could be met [e.g. 38]. In order to yield more thorough estimates, we took into account the structure of the population as a whole, weighted by gender and age, both in the historical and future calculations.

## Limitations and future investigations

One limitation of this study is that we make a simple assumption that food dietary requirements is a matter of calories and nutrients needed for the population, not considering actual patterns of

food consumption, neither food diversity nor food quality, but the domestic agricultural production. Nevertheless, the actual patterns of food consumption reflect income levels and food preferences hugely influenced by markets rather than what should be ideally consumed in terms of fresh and processed products [73]. Thus, considering the balance between production and demand of vegetables, cereals, meat and other food types could provide another approach for specific demand of land for each type of food product. For example, [75] considered the production and demand of vegetables at the municipality level in Brazil, providing information where vegetable sources are insufficient to meet regional needs. Further, we considered calories and selected macro- and micro-nutrients to which we had information. Edible vegetables which enhance the nutritional composition of the food that are consumed but not included in national surveys could not be considered in our analysis, although they have great importance for local food security. A full assessment of food and nutrition security will require consideration of such aspects. Given the importance of Brazil for the provision of key global and regional ecosystem services, the trade-offs between agricultural production and environmental impacts should be further investigated, especially in the scope of the planetary boundaries discussion.

It is important to mention that we do not address the actual trends in diets that are shifting towards greater consumption of meat and processed products, as well as the health issues associated with each type of diet. We did not include increases in national meat consumption in future scenarios because the average consumption is already above the levels recommended by the WHO. Also, because we did not find any evidence that increasing meat consumption could improve the sustainability of the agricultural sector [76, 77].

Further analysis and research are needed to address food waste, changes in agricultural technology,—(such as organic farms, crop fortification, GMO etc.), bioavailability of the nutrients, closing yield gaps and the impacts of climate change and water scarcity on yields [8, 23, 36, 37, 78]. Finally, we did not consider the role of the food imports in the land demand and diet scenarios. With more calories imported, less land is needed, though this means the environmental impacts of production are allocated to other countries [79, 80].

## Concluding remarks

Despite the limitations of the analysis, this work contributes to deepen the discussions about land use change due to agricultural expansion and its correlation to food security in Brazil. We provide evidence that the food and nutritional needs of the Brazilian population do not necessitate an expansion of agriculture land beyond its current range. We highlight that Brazil holds a unique condition for delivering food security, (addressed here strictly in terms of nutrient availability), while limiting its land use expansion and promoting sustainable intensification of food production. Policy enforcement against deforestation along with incentives for promoting biodiversity-friendly food production could significantly improve the sustainability of the Brazilian agricultural sector. Promoting more sustainable, healthy and diversified diets could alleviate further conversion of the vegetation cover for cattle ranching. This would allow key ecosystem services that are currently threatened by pastureland expansion to be maintained. Our methodology presents an improvement for further spatial analysis that include other indicators of nutrition, trade and environment. And the results presented could help the development of more effective public planning in Brazil towards sustainable land use, that considers the production and the consumption of adequate nutritious food.

## Supporting information

**S1 Table. Allocation of FBS's other uses and processed items.** The ratio of Oil and Ethanol production to "Processed" crop (e.g. soybeans or sugar cane) from FBS was used to convert

raw crop calories and nutrients to Oil or Ethanol. In the case of soybeans, the range of fraction of crop production that is converted to Oil is between 0.17 and 0.20 in the series, while the fraction of sugar cane ranges from 0.10 to 0.13. Once the "Processed" portions were calculated, they were allocated to "Export", "Feed" and "Food" according to the Oil and Sugar sheets.
(PDF)

**S2 Table. Daily nutritional requirements for the Brazilian population weighted by sex and age.** Data is shown for each 10 years, including future projections (2020–2060), to enhance visualization. RDA and AI values were taken from the United States Institute of Medicine and population projections from IBGE [46–48].
(PDF)

**S3 Table. Assumptions underlying the scenarios.**
(PDF)

**S4 Table. Percent of population whose energy and nutritional requirements were met in each scenario.** Data is shown for each 10 years, including future projections (2020–2060), to enhance visualization. The production of 2017 is the baseline for future projections, considering population dynamics and any agricultural expansion.
(PDF)

**S1 Fig. Percent of food calories and nutrients produced from pasturelands (cattle meat and milk).** Boxplots indicate the contribution of pasturelands (cattle meat and milk) for food calories and nutrients, in relation to the total food production in 30 years. In most years of the analysis, only calcium and vitamin A were not mostly produced from crops. Except for these nutrients, along with zinc and vitamin B12 (~30% to ~45% produced from pastures), more than 70% of the nutrients and calories are produced from food crops. In spite of that, pastures occupy most farmlands in Brazil.
(TIF)

## Acknowledgments

We are grateful for the valuable comments and suggestions made by Dr. James Gerber, as well as by the anonymous reviewers, which enhanced significantly the manuscript.

## Author Contributions

**Conceptualization:** João Pompeu, Jean Ometto.

**Data curation:** João Pompeu, Jacqueline Gerage.

**Formal analysis:** João Pompeu, Camille L. Nolasco.

**Funding acquisition:** Pete Smith, Jean Ometto.

**Investigation:** Camille L. Nolasco.

**Project administration:** Jean Ometto.

**Supervision:** Paul West, Pete Smith.

**Writing – original draft:** João Pompeu, Camille L. Nolasco.

**Writing – review & editing:** Camille L. Nolasco, Paul West, Pete Smith, Jean Ometto.

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
