## [Decision Letter · Decision Letter 0]

22 Jul 2020

PONE-D-20-13269

Is domestic agricultural production enough to meet national food security in Brazil?

PLOS ONE

Dear Dr. Pompeu,

Thank you for submitting your manuscript to PLOS ONE. After careful consideration, we feel that it has merit but does not fully meet PLOS ONE’s publication criteria as it currently stands. Therefore, we invite you to submit a revised version of the manuscript that addresses the points raised during the review process.

The comments from the reviewers indicate that substantial work needs to be done to the paper to get it in shape for publication. These comments should not be treated lightly.

We look forward to receiving your revised manuscript.

Kind regards,

Gideon Kruseman, Ph.D.

Academic Editor

PLOS ONE

Journal Requirements:

Additional Editor Comments (if provided):

The comments from the reviewers indicate that substantial work needs to be done to the paper to get it in shape for publication. These comments should not be treated lightly.

Reviewers' comments:

Reviewer's Responses to Questions

**Comments to the Author**

1. Is the manuscript technically sound, and do the data support the conclusions?

Reviewer #1: Partly

Reviewer #2: Partly

Reviewer #3: Partly

2. Has the statistical analysis been performed appropriately and rigorously? 

Reviewer #1: Yes

Reviewer #2: Yes

Reviewer #3: No

3. Have the authors made all data underlying the findings in their manuscript fully available?

Reviewer #1: Yes

Reviewer #2: No

Reviewer #3: No

4. Is the manuscript presented in an intelligible fashion and written in standard English?

Reviewer #1: Yes

Reviewer #2: No

Reviewer #3: Yes

5. Review Comments to the Author

Reviewer #1: General comments

The paper rightly notes the important tradeoff between agricultural production and environmental impacts, and provides a detailed and spatially disaggregated analysis of domestic production and nutrient availability in Brazil. However, the paper misses important distinctions between nutrient needs, food security, and food demand, as well as key drivers of the latter (including income and preferences). If the authors’ intent is to focus on the potential to reduce agriculture’s environmental footprint in Brazil through alternative diets (as seems to be the case), this should be made more explicit in the title. If, however, the intent is to assess food security (as suggested in the current title), much more attention is needed on income, consumer preferences, trade, and other economic factors.

Specific comments

Page 1, Title: food security has multiple dimensions (as the authors note on page 8), not all of which are examined in this paper, so a more accurate title would be “Is domestic agricultural production enough to meet national nutrient needs in Brazil?” (or “How much would alternative diets reduce agriculture’s environmental footprint in Brazil?”). At a minimum, the authors should note earlier in the paper that they focus on nutrient availability, and are using the term food security in that restricted sense.

Page 1, Abstract: the paper’s main question is posed in terms of environment (land) and food security, but needs better motivation for the authors’ focus on domestic production.

Page 2, Line 13: trade is acknowledged as important for global food security, i.e. not only for Brazil but for countries that import from Brazil. But export earnings are not among the outcomes considered later in the paper alongside the environmental footprint of domestic production.

P2, L25: need to distinguish local/national environmental impacts (e.g. soil erosion, water quality) vs those with global implications (e.g. greenhouse gas emissions). The asserted link between land use change and negative impacts on agricultural yields also needs further explanation and support.

P2, L29: clarify expansion: does this refer to an increase in crop area, or production, or demand?

P2, L48: does food just mean calories, or also micronutrients? (clarified on the next page)

P3, L65: need to support the statement that pasture has "marginal contribution to nutrient production".

P4, L134: R2 over different time periods seems like a weak basis for model selection.

P7, L228: it’s problematic to say that crop calories increased threefold, animal calories increased fourfold, but this had minor effects on food availability. It may have had a minor effect on calories consumed domestically (which is largely a function of domestic population), but that is not the same as food availability.

P7, L241: the fact that Brazil exports 2/3 of the calories it produces makes it clear that domestic production capacity is not the constraining factor for domestic nutrient availability – as the authors note on P8, L280, and which should not be a surprise to most readers.

P8, L269: the fact that there is no change in number of people fed (in terms of calories) reinforces the point that calorie consumption is closely correlated with population, and signals that consumers switch to higher value foods (in terms of price and nutrients) as their incomes rise.

P8, L280: “food production was not the major issue for food security in Brazil”, neither in the future… At L296 the authors correctly note that food security in Brazil is a matter of access, stability and utilization – which, it should be noted, are not addressed in this paper.

P8, L301: are projected needs based only on changes in population, i.e. no change in income, prices, preferences, productivity, etc? If these important factors are left out, this needs to be noted explicitly, with a discussion of what difference that makes to the results.

P9, L321: was maintaining trade mentioned earlier as a specific goal/constraint? Also note “feed its population” here seems to mean (just) providing sufficient calories and micronutrients (acknowledged on P11, L424).

P11, L387: diets shifting towards more meat: need to consider why (e.g. incomes and preferences), and what that means for measures to shift diets towards less meat.

P11, L399: also reducing losses and waste.

P11, L429: not clear.

P12, L454: demand not the same as needs (also food security in L455).

Reviewer #2: This paper analyses agricultural land-use in Brazil by mapping land use for agricultural exports, comparing food production to nutritional requirements, and calculating scenarios for hypothetical diet changes. The paper is separable into two parts: the mapping of land-use and a comparison of agricultural land use with nutritional requirements. Both parts of the paper are based on appropriate data and provide the basis for a relevant and scientifically sound analysis. However, both parts would substantially profit from further refinements as described below. For the mapping exercise the current methodology is appropriate and does not require changes. The comparison of land and nutrient requirements needs to address methodological concerns, especially regarding the substitution between different agricultural commodities. As the interpretation of the two parts is not dependent on each other, I suggest that the authors split the work into two separate articles and resubmit the new papers. This would make better use of the large amount of work that appears to have gone into the paper and the scientific and policy potential visible in it.

In the first part, the authors use a rich dataset of municipality level agricultural production information and combine this with satellite data to produce a detailed map of land use for the production of different agricultural commodities. This detailed data is used to create a map of exported calorie shares, showing where most land is used for exports. This part of the paper makes good use of the detailed data. The authors do not analyze the map further and do not compare it with other geographical distributions. It would be very interesting to a) compare this map with maps of biodiversity or other measures showing the potential damage of land use changes and b) compare it to other relevant indicators like the economic value of the products. Comparing between these different layers of the map would help to show the trade-offs that certain agricultural policies require. This would give the work of the authors a much higher policy relevance.

The second part of the paper does not use the geographical details of the data but uses national aggregates of nutrient production and agricultural land use. The authors compare food production with the nutrient requirements of the population and apply different scenarios to show how much of the area would be required to meet the requirements. The authors differentiate between calorie requirements and a wider set of macro and micronutrient requirements. Using demographic projections, the authors show historic and future land requirements under the different scenarios. The topic and general approach of this part is interesting but it requires some alterations to the approach to provide practical relevance. My main concern is that the authors assume no change in the land use and the crops grown when shifting to a different diet. This would mean that consumers substitute meat consumption with the direct consumption of the respective feeds. Beyond being a very unlikely scenario, this has strong implications for the results of the study. E.g. the current paper concludes that the production system would not provide sufficient vitamin A under a vegan diet. A substitution of livestock products with fruits and vegetables instead of feed crops would most likely address this. Further, the authors leave the topic of agricultural exports unaffected. This is a reasonable simplification but it would be good to clarify this a bit more. Additionally, the methodology used to set nutrient requirements could be strengthened. The authors should shortly explain why they use the recommended dietary intake as reference in their calculation of micronutrient requirements while population level analyses of nutrient adequacy usually rely on estimated average requirements. Lastly, the requirements for all nutrients but calories are calculated from the demographic structure. The authors could do the same for calories, thereby harmonizing the approach.

Minor comments:

1) The authors should provide supporting arguments for the assumption that all cattle is grazing (line 121) or adapt the calculation. Especially for dairy production, single reliance on grazing appears questionable.

2) The authors could consider to introduce the quoted inequality in the food distribution into their calculation of required nutrients. This would mean to calculate requirements under the assumption that the current distribution remains unchanged. This suggestion is optional and authors might choose the current simplification.

3) Formula (1) contains XY as index but the explanation does not explain what these are. Also, the term fcrop is described to be “the map of a certain crop area”, which is unclear. I assume that this might refer to the state of the specific grid cell. I suggest to fully review the formula and its description.

4) Figure 1 and/or its description are unclear. E.g. It is unclear if the dotted and dashed lines are part of the solid lines and why animal and crop calories proportions in some years sum to more than 100% of all produced calories and in other years to less.

5) Replace the term “food system” with “domestic demand” or a similar turn to avoid confusion with the food systems literature (e.g. line 244).

6) The description of figure 2 refers to the graphs in the wrong direction.

7) Line 296f. refers to food stability. It is unclear what this term means.

8) Line 306 appears to use “with” instead of “without”.

9) In line 403ff. it is unclear what kind of estimates this refers to and why no clear number is given (“circa”).

10) Lines 415 ff. refers to income deciles representing “less than 1% of the population” while a decile per definition is 10% percent of the population.

11) I suggest to carefully review the writing and style: At a few points punctuation (e.g. in the abstract), letters, or words (e.g. line 3) are missing or too much (e.g. line 205, line 273, caption of figure 3, line 312). In line 47 the term ”on the other hand” is used but the sentences do not describe contradictory arguments. The paragraph from line 395 repeats the term required five times, where it would read better with the usage of other terms. The sentence in lines 431ff. is incomplete.

12) Line 65 describes the nutrient supply from beef as marginal. Based on the large quantity of specific nutrients (e.g. vitamin B 12) produced for export and domestic use (as visible in the authors’ own calculation) this seems not to be fitting. It would be better to make an efficiency argument regarding the nutrients produced per hectare. I further recommend to split the sentence.

13) The link on reference 42 does not work.

14) Table S1 contains a group of crops, where it is unclear why these are selected. To make the table more useful it would be good if the authors could tabulate the numeric values, e.g. for each of the three mapped periods per crop or crop group.

15) In table S2 some units are wrong (e.g. gram of vitamin C per day).

16) In table S4 it would be interesting to know if aquaculture is excluded. Also, the description refers to agricultural expansion while the paper otherwise assumes that the land usage for each crop remains stable for this form of analysis.

17) The uploaded data contains the nutrients per municipality but not the information required to reproduced the map for exports or the land usage per grid cell. This would not allow to reproduce the results.

Reviewer #3: The objective of the reviewed article is ostensibly to understand the capacity of the Brazilian agriculture system to provide for food security at the country/national level. The basic question, it seems, is to understand whether or not there is sufficient capacity in the Brazilian agriculture system to feed the population now and well into the future. The premise is reasonable and there are findings presented in the paper that could support decision making related to food system policy.

While there are some novel elements to the paper, there are a number of issues - some significant, some modest - that need to be addressed before the paper is ready to be published.

The first issue of concern is simply the core thesis of the paper. Is it that Brazil has sufficient agricultural capacity to provide for food security, or lower levels of meat consumption are the required in order for Brazil to achieve food security? I ask this as the the former seems to get obfuscated by the latter in several places in the paper. Likewise, related to the same, there have been a surge of recent papers and reports (the EAT Lancet Report, for example) that have unpacked similar issues at the global scale. There is a lot of potential context around the relationship between food security and planetary boundaries that could be considered in relation to the present effort, but much of this literature is omitted from the current manuscript. Another area of concern is the authors are emphasizing food security in general terms but really only address the issue of food availability (at least in relation to the problem statement - other issues come out in the discussion, but this again leaves me wondering about the key messages and question of this work).

A second area for consideration is in relation to the relatively complex dynamics that happen with regard to prices, land use, trade, etc., as the structure of agriculture production shifts. The authors make a number of simplifying assumptions (many of them reasonable, I might add), but largely treat the question of agriculture production as an economic isolated construct. From a food system perspective, it is critical to consider the different trade-offs that may come with different policy decisions (e.g., perhaps policy to reduce meat consumption). Aspects of this are loosely acknowledged in the MS, but only generally (e.g., lowering the Brazilian food footprint while maintaining trade in animal products); that said, these issues are highly complex and merit additional consideration given the way in which the argumentation is made in the paper.

The issues mentioned above also have significant implications for the future scenarios. There are several scenarios discussed relating to changing diets (specifically towards lower meat consumption) and land use. As mentioned above with regard to the central thesis, the precise justification of the scenarios does not come across as totally clear. Is this an arbitrary choice, is this related to a policy objectives, is this consistent with the recent planetary boundaries work, etc?

While the inputs are clear and certain key aspects of the methods are clear (e.g., the specific periods associated with the projections), the methods as a whole are unclear in terms of the whole and how each step/part links to the central thesis. One relatively clear example (at least for this reviewer) is the use of the MapBiomass data. There seems to be a fairly significant disconnect between the highly detailed MapBiomass analysis and the later projections forward in time. Again, this might connect back to the needed to more crisply define the objective and specific research question associated with the manuscript.

There are some interesting results and the authors should consider how they help focus the central premise of the paper. For example, they talk about national caloric production and then growth of population. Would this be better expressed in per capita terms given the emphasis on food security?

Finally, I am not sure if the confidence analysis adds much support. Part of this stems from the lack of clarity on certain aspects of the analysis. Given the extrapolative nature of the study, it seems there should either be systematic characterization of each of the analytic steps for verification and validation and/or some sort of sensitivity analysis associated with the overall model results.

While challenging, none of the above are insurmountable, and I would therefore recommend major revisions and a substantial clarification on both the thesis and its basis before the paper is ready for publication.

There are also a number of minor errors and style issues in the document and the manuscript would benefit from careful proof reading in advance of resubmission. For example, please see:

Line 61 - Our study provideS...

Line 153 - nutrinional - should be nutritional

6. PLOS authors have the option to publish the peer review history of their article (what does this mean?). If published, this will include your full peer review and any attached files.

Reviewer #1: No

Reviewer #2: No

Reviewer #3: No

---

## [Author Response · Author response to Decision Letter 0]

22 Jan 2021

RESPONSE TO REVIEWERS

Reviewer #1 

General comments

The paper rightly notes the important tradeoff between agricultural production and environmental impacts, and provides a detailed and spatially disaggregated analysis of domestic production and nutrient availability in Brazil. However, the paper misses important distinctions between nutrient needs, food security, and food demand, as well as key drivers of the latter (including income and preferences). If the authors’ intent is to focus on the potential to reduce agriculture’s environmental footprint in Brazil through alternative diets (as seems to be the case), this should be made more explicit in the title. If, however, the intent is to assess food security (as suggested in the current title), much more attention is needed on income, consumer preferences, trade, and other economic factors.

Response: Thanks for your careful evaluation of our work. We do agree with the reviewer that a complete assessment of food security requires addressing the broader aspects of food processing, retailing and, more importantly, access to healthy food. We thus changed the title and considered the suggestions made by the reviewer, as follows in the specific comments. 

Specific comments

-Page 1, Title: food security has multiple dimensions (as the authors note on page 8), not all of which are examined in this paper, so a more accurate title would be “Is domestic agricultural production enough to meet national nutrient needs in Brazil?” (or “How much would alternative diets reduce agriculture’s environmental footprint in Brazil?”). At a minimum, the authors should note earlier in the paper that they focus on nutrient availability, and are using the term food security in that restricted sense.

Response: We changed the title to “Is domestic agricultural production sufficient to meet national food nutrient needs in Brazil?” and highlighted in the Introduction section the focus on food nutrients availability at these parts:

“This resulted in increasing deforestation trends since 2012, reaching in 2020 its highest peak since 2008 [27–29]. Political efforts to reduce environmental protection in Brazil were often based on the argument that more land is needed to produce food for the Brazilian population [30-32]. This argument was central to the debate that changed the environmental legislation in 2012 and is still argued in some proposals, both from the national congress and government, for promoting the agricultural expansion [30]. However, the proportion of land needed in the territory for meeting the nutritional food demand of the Brazilian population, currently around 210 million people, is unknown.”

“Shifting diets towards those which contain lower quantities of livestock products could reduce the amount of land needed to produce food.”

-Page 1, Abstract: the paper’s main question is posed in terms of environment (land) and food security, but needs better motivation for the authors’ focus on domestic production.

Response: We have rewritten the abstract 

-Page 2, Line 13: trade is acknowledged as important for global food security, i.e. not only for Brazil but for countries that import from Brazil. But export earnings are not among the outcomes considered later in the paper alongside the environmental footprint of domestic production.

Response: Addressed in the discussion section.

“In this sense, [79] showed that agricultural expansion for export could improve the development indexes due to income and labour generation mainly in the Brazilian municipalities at the agricultural frontiers. Also, [80] claim that the expansion and improvement of this sector has generated substantial improvements in well-being throughout Brazil in several ways, such as governmental revenues for welfare programs and increasing private purchasing power. However, both works acknowledge that this is accompanied by income and land concentration, displacement of peasants and rural to urban migration with consequent losses of livelihoods when people move to marginal conditions. While the earnings from commodity exports improve local food security by increasing the economic affordability of food to more employed families, the benefits of global trade do not always necessarily overcome the social impacts. A socially-inclusive strategy and better multisectoral planning is needed for reducing exclusion in agricultural consolidated and under expansion, areas.”

-P2, L25: need to distinguish local/national environmental impacts (e.g. soil erosion, water quality) vs those with global implications (e.g. greenhouse gas emissions). The asserted link between land use change and negative impacts on agricultural yields also needs further explanation and support.

Response: Addressed in the text.

“Thus, additional large-scale land use changes are likely to cause negative environmental impacts with global implications for climate change via GHG emissions. Soil erosion, nutrient loss, water depletion and biodiversity loss [25, 74] are important regional impacts of land use changes. Agricultural yields are also affected by these impacts, increasing the demand for more land for production, leading to the positive feedback [19]. As already demonstrated in Brazil, extensive land clearing for agriculture in the Cerrado and eastern Amazon alters regional weather, reducing maize yields by 6% to 8%, undermining the intended use of the cleared land [75].”

-P2, L29: clarify expansion: does this refer to an increase in crop area, or production, or demand?

Response: “Land use expansion for growing commodity crops...”

-P2, L48: does food just mean calories, or also micronutrients? (clarified on the next page)

Response: “Shifting diets towards those that contain lower quantities of livestock products could reduce the amount of land needed to produce food in terms of calories, macro- and micro-nutrients for human consumption.”

-P3, L65: need to support the statement that pasture has "marginal contribution to nutrient production".

Response: We clarified the sentence, added the following statement in the results section to show that most food nutrients are produced from crops and added the new supplementary file 5.

“Calcium and vitamin A for human consumption are mostly cattle based (up to 75% in certain years, as shown S1 Fig). Except for these nutrients, along with zinc and vitamin B12 (~30% to ~45% produced from pastures), more than 70% of the nutrients and calories are produced from food crops (S1 Fig).”

-P4, L134: R2 over different time periods seems like a weak basis for model selection.

Response: We used the adjusted R-squared to address the issue of overfitting when using R-squared with different number of variables. This was the case, in which we tested the period with better correlation in order to minimize the effect of abrupt changes in the trends over time.

-P7, L228: it’s problematic to say that crop calories increased threefold, animal calories increased fourfold, but this had minor effects on food availability. It may have had a minor effect on calories consumed domestically (which is largely a function of domestic population), but that is not the same as food availability.

Response: We do agree that the way we wrote led the authors to a misinterpretation of what we meant and made it clearer in the text now. 

“Such an increase in crop production was mainly driven by export commodities with a less pronounced increase in food crops. The amount of food crop calories increased by 19%, from 1.40 x 1014 to 1.67 x 1014 kcal, while its proportion in total production declined from 32.8% to 13.0% (Fig 1).”

-P7, L241: the fact that Brazil exports 2/3 of the calories it produces makes it clear that domestic production capacity is not the constraining factor for domestic nutrient availability – as the authors note on P8, L280, and which should not be a surprise to most readers.

Response: This reinforces what is discussed later in the text that food insecurity is a matter of food access and that land use changes are not necessary to meet the nutritional requirements if food is properly distributed.

-P8, L269: the fact that there is no change in number of people fed (in terms of calories) reinforces the point that calorie consumption is closely correlated with population, and signals that consumers switch to higher value foods (in terms of price and nutrients) as their incomes rise.

Response: What is shown in this argument is that food production increases as population also increases and not that people are switching to more caloric diets. We clarified in the text:

“This provides evidence that the agricultural sector as a whole is shifting to market-oriented commodity production while the food calorie increases at the same rate of the population.”

-P8, L280: “food production was not the major issue for food security in Brazil”, neither in the future… At L296 the authors correctly note that food security in Brazil is a matter of access, stability and utilization – which, it should be noted, are not addressed in this paper.

Response: We strictly focused on the production side of food security, as food availability is the first constraint for food security. We clarified our focus with a revised title and in the introduction section:

“Focusing more specifically on the production side of food security (also referred to as food availability in food security literature), we combined supply and demand for calories, macro- and micro-nutrients based on three decades of data on agricultural production and trade, land use and nutritional requirements of the population.”

-P8, L301: are projected needs based only on changes in population, i.e. no change in income, prices, preferences, productivity, etc? If these important factors are left out, this needs to be noted explicitly, with a discussion of what difference that makes to the results.

Response: Addressed in the methods and in the discussion sections, respectively: 

“Caloric and nutrient demand were calculated on the basis of the population projections and did not consider changes in income or individual preferences.”

“Moreover, rapid increase in food prices without increase in income or purchasing power is left out due to the limited scope of the analysis. However, our results indicate that proper food policies to buffer these side effects on food security are essential, once, as noted above, food production should not be a constraining factor for achieving food security in Brazil.”

-P9, L321: was maintaining trade mentioned earlier as a specific goal/constraint? Also note “feed its population” here seems to mean (just) providing sufficient calories and micronutrients (acknowledged on P11, L424).

Response: International trade was not the focus of the work, so we meant that even using 10% less land, enough food nutrients were produced even excluding the exported nutrients. 

“In the past, Brazil could have used 10% less land to produce enough calories and nutrients for domestic supply (BAU). If food was properly distributed, even using less land for food production, would not compromise Brazilian agricultural exports.”

-P11, L387: diets shifting towards more meat: need to consider why (e.g. incomes and preferences), and what that means for measures to shift diets towards less meat.

Response: Acknowledged in the Discussion:

“The analysis of [63] reveals that income plays an important role in this pattern, not because of individual preferences to consume more meat, but due to the affordability of animal sources of protein. Strengthening food security policies that ensure the access to more diversified and fresh food could play an important role in shifting diets towards less meat consumption.”

-P11, L399: also reducing losses and waste.

Response: Addressed in the text. 

-P11, L429: not clear.

Response: “Thus, considering the balance between production and demand of vegetables, cereals, meat and other food types could provide another approach of specific demand of land for each type of food product. For example, [68] considered the production and demand of vegetables at the municipality level in Brazil, providing information where vegetable sources are insufficient to meet regional needs.”

-P12, L454: demand not the same as needs (also food security in L455).

Response: “We highlight that Brazil holds a unique condition for delivering food security, (addressed here strictly in terms of nutrient availability), while limiting its land use expansion and promoting sustainable intensification of food production.”

Reviewer #2

This paper analyses agricultural land-use in Brazil by mapping land use for agricultural exports, comparing food production to nutritional requirements, and calculating scenarios for hypothetical diet changes. The paper is separable into two parts: the mapping of land-use and a comparison of agricultural land use with nutritional requirements. Both parts of the paper are based on appropriate data and provide the basis for a relevant and scientifically sound analysis. However, both parts would substantially profit from further refinements as described below. For the mapping exercise the current methodology is appropriate and does not require changes. The comparison of land and nutrient requirements needs to address methodological concerns, especially regarding the substitution between different agricultural commodities. As the interpretation of the two parts is not dependent on each other, I suggest that the authors split the work into two separate articles and resubmit the new papers. This would make better use of the large amount of work that appears to have gone into the paper and the scientific and policy potential visible in it.

In the first part, the authors use a rich dataset of municipality level agricultural production information and combine this with satellite data to produce a detailed map of land use for the production of different agricultural commodities. This detailed data is used to create a map of exported calorie shares, showing where most land is used for exports. This part of the paper makes good use of the detailed data. The authors do not analyze the map further and do not compare it with other geographical distributions. It would be very interesting to a) compare this map with maps of biodiversity or other measures showing the potential damage of land use changes and b) compare it to other relevant indicators like the economic value of the products. Comparing between these different layers of the map would help to show the trade-offs that certain agricultural policies require. This would give the work of the authors a much higher policy relevance.

The second part of the paper does not use the geographical details of the data but uses national aggregates of nutrient production and agricultural land use. The authors compare food production with the nutrient requirements of the population and apply different scenarios to show how much of the area would be required to meet the requirements. The authors differentiate between calorie requirements and a wider set of macro and micronutrient requirements. Using demographic projections, the authors show historic and future land requirements under the different scenarios. The topic and general approach of this part is interesting but it requires some alterations to the approach to provide practical relevance. My main concern is that the authors assume no change in the land use and the crops grown when shifting to a different diet. This would mean that consumers substitute meat consumption with the direct consumption of the respective feeds. Beyond being a very unlikely scenario, this has strong implications for the results of the study. E.g. the current paper concludes that the production system would not provide sufficient vitamin A under a vegan diet. A substitution of livestock products with fruits and vegetables instead of feed crops would most likely address this. Further, the authors leave the topic of agricultural exports unaffected. This is a reasonable simplification but it would be good to clarify this a bit more. Additionally, the methodology used to set nutrient requirements could be strengthened. The authors should shortly explain why they use the recommended dietary intake as reference in their calculation of micronutrient requirements while population level analyses of nutrient adequacy usually rely on estimated average requirements. Lastly, the requirements for all nutrients but calories are calculated from the demographic structure. The authors could do the same for calories, thereby harmonizing the approach.

Response: We are grateful for the valuable considerations to enhance the work we have done. 

After a careful evaluation, we decided to remove the spatially explicit analysis from our results. We understand the suggestions of the reviewer and agree that we could make a better discussion of the spatial patterns of food and agricultural commodity production in Brazil. As this is not the main scope of the present manuscript, we made the following changes:

-Made clear in the methods that MapBiomas was used to retrieve yearly pastureland area and removed the subsection “Mapping food and exports”, which was strictly related to the maps:

“Additionally, we obtained crop and pastureland areas for every year from 1988 to 2017 from MapBiomas’s land use and land cover maps (collection 3.0). MapBiomas is a partnership of private companies, universities and NGOs that use all available series of remotely sensed images from Landsat satellites to produce land use and land cover maps for the Brazilian territory, in the period of 1985 to 2017, with 30 metres of spatial resolution [39]. Based on the municipal data from PAM, the class “Crop or Pasture”, an error class in MapBiomas that cannot be distinguished due to spectral confusion, was allocated to “Pasturelands” or “Croplands” classes, adjusting the cropland maps to the official Brazilian cropland area. This allowed us a yearly estimate of the pasture area at the national level.”

-Removed the maps (Figure 2) and the following paragraph: 

“Currently, most of the crop calories that are produced in the north-east and in the extreme south regions of the country are produced as food (Fig 2), while those produced in the centre-south are exported. A similar pattern, with less pronounced division, was observed for 2003, when Brazil consolidated as a net exporter of crops [48]. In the beginning of the study series, 1988, a very different distribution was found, when higher proportions of national crop production were used for food (Fig 2).”

After these changes, we considered the commentaries about the argued “second part”. We highlighted in the Introduction that land use expansion for food production is often argued as necessary in policies to weaken environmental protection in Brazil. The basis of our work is to investigate whether the current land use could provide sufficient nutrients for the Brazilian population. This is the main reason to assume no change in the land use. 

“the current paper concludes that the production system would not provide sufficient vitamin A under a vegan diet. A substitution of livestock products with fruits and vegetables instead of feed crops would most likely address this.”

We conclude that the current production system would fail in provide sufficient vitamin A under a vegan diet. This is why we recognize that “A full assessment of food and nutrition security will require consideration of such aspects”, which is at the scope of our future investigations. 

Regarding exports, we left unchanged because our focus is on the domestic food production. Our results show that even without changes in the exports, less land could be used to achieve the nutritional demands of the population.

Finally, we added the following sentences in the methods to clarify the confusion about RDA and caloric calculations:

“According to the United States Institute of Medicine, there is no RDA for energy because energy intakes above the Estimated Energy Requirement would be expected to result in weight gain [42]. Thus, the proportion of the population potentially fed with national food production was estimated on the basis of 2,450 kcal/person/day diets (894,250 kcal/yr), which is the maximum value of Average Dietary Energy Requirements (ADER) recommended by FAO for Brazil.”

[…]

“Following the methods from [50], whenever available, we use the Recommended Dietary Allowance (RDA) which represents the average intake that would meet the needs of 97.5% of healthy individuals in a group. If an RDA was not available for certain nutrients, we used Adequate Intake (AI), which is the level of intake assumed to be adequate for healthy individuals. The daily requirements for each nutrient in each year, weighted for the Brazilian population, is found in the S2 Table. This way, we had both nutrients and calories averaged for the structure of the Brazilian population every year in the period of analysis. Caloric and nutrient demand were calculated on the basis of the population projections and did not consider changes in income or individual preferences.”

Minor comments:

1) The authors should provide supporting arguments for the assumption that all cattle is grazing (line 121) or adapt the calculation. Especially for dairy production, single reliance on grazing appears questionable.

Response: We apologize and recognize that this sentence leads to a misinterpretation of our assumptions. We did not mean that dairy production relies only on grazing, as clarified in the line 176-177 of the first version of the submitted manuscript. We meant that we did not account for dairy under confinement systems, that corresponds to only 2.4% of the production in Brazil. Thanks for the advice.

“We assumed that both beef and milk cattle were produced in grazing and not confinement systems, which accounts for only a small fraction (2.4%) of cattle production in Brazil [77].”

2) The authors could consider to introduce the quoted inequality in the food distribution into their calculation of required nutrients. This would mean to calculate requirements under the assumption that the current distribution remains unchanged. This suggestion is optional and authors might choose the current simplification.

Response: We made a brief statement in the Discussion to address the important issue of inequality:

“The causes of food insecurity in Brazil are likely to be rooted in poverty and in the inefficient distribution and retailing of food, that constrains stable access to healthy food. Between 2013 and 2018, mild food insecurity increased by 71.5%, while severe, which characterizes hunger, increased by 48.8%, totalling around 10.3 million people in food insecurity in Brazil [78]. Even while exporting more agricultural commodities and producing enough food, Brazil is in danger of again being reinstated on the world hunger map.”

3) Formula (1) contains XY as index but the explanation does not explain what these are. Also, the term fcrop is described to be “the map of a certain crop area”, which is unclear. I assume that this might refer to the state of the specific grid cell. I suggest to fully review the formula and its description.

Response: Excluded from the manuscript as part of the spatially explicit analysis.

4) Figure 1 and/or its description are unclear. E.g. It is unclear if the dotted and dashed lines are part of the solid lines and why animal and crop calories proportions in some years sum to more than 100% of all produced calories and in other years to less.

Response: Clarified in the description.

“Caloric production in Brazil from 1988 to 2017. Animal (red) and crop (green) calories sum 100% each, disaggregated into calories for export (dashed lines), domestic food consumption (solid lines) and animal feed (dotted lines).”

5) Replace the term “food system” with “domestic demand” or a similar turn to avoid confusion with the food systems literature (e.g. line 244).

Response: Addressed in the text.

6) The description of figure 2 refers to the graphs in the wrong direction.

Response: Excluded from the manuscript as part of the spatially explicit analysis.

7) Line 296f. refers to food stability. It is unclear what this term means.

Response: Stability is how food is produced, accessed and used in time. We found that the word is redundant in the sentence and removed it. 

8) Line 306 appears to use “with” instead of “without”.

Response: Addressed in the text. 

9) In line 403ff. it is unclear what kind of estimates this refers to and why no clear number is given (“circa”).

Response: Clarified in the text.

“based on estimates of circa the year 2000 (1997-2003).”

10) Lines 415 ff. refers to income deciles representing “less than 1% of the population” while a decile per definition is 10% percent of the population.

Response: This refers to income deciles instead of population deciles. In this case, 1% of the population account for the 10% higher income. 

11) I suggest to carefully review the writing and style: At a few points punctuation (e.g. in the abstract), letters, or words (e.g. line 3) are missing or too much (e.g. line 205, line 273, caption of figure 3, line 312). In line 47 the term ”on the other hand” is used but the sentences do not describe contradictory arguments. The paragraph from line 395 repeats the term required five times, where it would read better with the usage of other terms. The sentence in lines 431ff. is incomplete.

Response: Thanks for the advice. We have corrected typos and checked for minor spelling errors. 

12) Line 65 describes the nutrient supply from beef as marginal. Based on the large quantity of specific nutrients (e.g. vitamin B 12) produced for export and domestic use (as visible in the authors’ own calculation) this seems not to be fitting. It would be better to make an efficiency argument regarding the nutrients produced per hectare. I further recommend to split the sentence.

Response: We clarified the sentence, added the following statement in the results section to show that most food nutrients are produced from crops and added the new supplementary file 5.

“Calcium and vitamin A for human consumption are mostly produced in pastures (up to 75% in certain years, as shown S1 Fig). Except for these nutrients, along with zinc and vitamin B12 (~30% to ~45% produced from pastures), more than 70% of the nutrients and calories are produced from food crops (S1 Fig).”

13) The link on reference 42 does not work.

Response: The following link is corrected:

https://www.nal.usda.gov/sites/default/files/fnic_uploads/Framework_DRI_Development.pdf

14) Table S1 contains a group of crops, where it is unclear why these are selected. To make the table more useful it would be good if the authors could tabulate the numeric values, e.g. for each of the three mapped periods per crop or crop group.

Response: Clarified why only a few crops are listed: 

“Crops omitted from this table were allocated exactly as reported in the FBS for each specific crop.”

15) In table S2 some units are wrong (e.g. gram of vitamin C per day).

Response: Thanks for the advice. We checked all the units and corrected this wrong one. 

16) In table S4 it would be interesting to know if aquaculture is excluded. Also, the description refers to agricultural expansion while the paper otherwise assumes that the land usage for each crop remains stable for this form of analysis.

Response: The major land use in Brazil is agriculture, especially for livestock. According to the Agricultural Census of 2017 (Table 6778), only 0.37% of the farms had any form of aquaculture and this is why we did not include it in our analysis.

17) The uploaded data contains the nutrients per municipality but not the information required to reproduced the map for exports or the land usage per grid cell. This would not allow to reproduce the results.

Response: Excluded from the manuscript as part of the spatially explicit analysis.

Reviewer #3

The objective of the reviewed article is ostensibly to understand the capacity of the Brazilian agriculture system to provide for food security at the country/national level. The basic question, it seems, is to understand whether or not there is sufficient capacity in the Brazilian agriculture system to feed the population now and well into the future. The premise is reasonable and there are findings presented in the paper that could support decision making related to food system policy.

While there are some novel elements to the paper, there are a number of issues - some significant, some modest - that need to be addressed before the paper is ready to be published.

The first issue of concern is simply the core thesis of the paper. Is it that Brazil has sufficient agricultural capacity to provide for food security, or lower levels of meat consumption are the required in order for Brazil to achieve food security? I ask this as the the former seems to get obfuscated by the latter in several places in the paper. Likewise, related to the same, there have been a surge of recent papers and reports (the EAT Lancet Report, for example) that have unpacked similar issues at the global scale. There is a lot of potential context around the relationship between food security and planetary boundaries that could be considered in relation to the present effort, but much of this literature is omitted from the current manuscript. Another area of concern is the authors are emphasizing food security in general terms but really only address the issue of food availability (at least in relation to the problem statement - other issues come out in the discussion, but this again leaves me wondering about the key messages and question of this work).

A second area for consideration is in relation to the relatively complex dynamics that happen with regard to prices, land use, trade, etc., as the structure of agriculture production shifts. The authors make a number of simplifying assumptions (many of them reasonable, I might add), but largely treat the question of agriculture production as an economic isolated construct. From a food system perspective, it is critical to consider the different trade-offs that may come with different policy decisions (e.g., perhaps policy to reduce meat consumption). Aspects of this are loosely acknowledged in the MS, but only generally (e.g., lowering the Brazilian food footprint while maintaining trade in animal products); that said, these issues are highly complex and merit additional consideration given the way in which the argumentation is made in the paper.

The issues mentioned above also have significant implications for the future scenarios. There are several scenarios discussed relating to changing diets (specifically towards lower meat consumption) and land use. As mentioned above with regard to the central thesis, the precise justification of the scenarios does not come across as totally clear. Is this an arbitrary choice, is this related to a policy objectives, is this consistent with the recent planetary boundaries work, etc?

While the inputs are clear and certain key aspects of the methods are clear (e.g., the specific periods associated with the projections), the methods as a whole are unclear in terms of the whole and how each step/part links to the central thesis. One relatively clear example (at least for this reviewer) is the use of the MapBiomass data. There seems to be a fairly significant disconnect between the highly detailed MapBiomass analysis and the later projections forward in time. Again, this might connect back to the needed to more crisply define the objective and specific research question associated with the manuscript.

There are some interesting results and the authors should consider how they help focus the central premise of the paper. For example, they talk about national caloric production and then growth of population. Would this be better expressed in per capita terms given the emphasis on food security?

Finally, I am not sure if the confidence analysis adds much support. Part of this stems from the lack of clarity on certain aspects of the analysis. Given the extrapolative nature of the study, it seems there should either be systematic characterization of each of the analytic steps for verification and validation and/or some sort of sensitivity analysis associated with the overall model results.

While challenging, none of the above are insurmountable, and I would therefore recommend major revisions and a substantial clarification on both the thesis and its basis before the paper is ready for publication.

Response: We thank the reviewer for the contribution to improving the manuscript. In general, the major comments were addressed in the responses to the other reviewers. Some specific topics related to the comments are the following:

- We improved the introduction to show that our aim is to explore if the current land use, along with different dietary patterns, would be enough or more land is needed to produce more food. The need for agricultural expansion is often claimed by policymakers in Brazil and our results point in the opposite direction. Neither dietary nor land use changes are needed to supply the food nutrients for the population. In addition, dietary changes could save a considerable amount of land, also pointing to the fact that no agricultural land use change is needed in Brazil. We recognize that diets should contain large amounts of fresh and diversified food, but this is not at the scope of this analysis. 

At this point, the highlighted parts were added to the main text:

“Despite law enforcement to reduce deforestation in Brazil from 2004 to 2012, significant changes in the environmental legislation were made under pressure from the agribusiness sector [30]. At the same time, the government reduced its public budget for environmental actions and benefited agribusiness expansion with financial support. This resulted in increasing deforestation trends since 2012, reaching in 2020 its highest peak since 2008 [27–29]. 

Political efforts to reduce environmental protection in Brazil were often based on the argument that more land is needed to produce food to the Brazilian population [30-32]. This argument was central in the debate that changed the environmental legislation in 2012 and is still argued in some proposals, both from the national congress and government, for promoting the agricultural expansion [30]. However, the proportion of land needed in the territory for meeting the nutritional food demand of the Brazilian population, currently around 210 million people, is unknown. [31–33] argue that increasing the productivity of pasturelands could negate the need for any further land to be deforested land for food production. Additionally, [34–36], point to the importance of considering that more food can be delivered on the same amount of land if dietary preferences are changed. Enriching diets with plant-based foods and with fewer animal sources is argued to be beneficial for both human health and for the ecosystems [76], reducing the environmental pressures of food production on the environment. Shifting diets towards those which contain lower quantities of livestock products could reduce the amount of land needed to produce food, in terms of calories, macro- and micro-nutrients for human consumption. Taken together, increasing yields through sustainable intensification and changing diets could make a significant contribution to deliver food security while reducing the pressure on the environment due to land use expansion.”

- We focused on the land use as deforestation for agricultural expansion is currently one of the key environmental issues in Brazil. A full assessment of trade-offs also requires broader evaluation of ecosystem services which is not our main objective here.

“Given the importance of Brazil for the provision of key global and regional ecosystem services, the trade-offs between agricultural production and environmental impacts should be further investigated, specially in the scope of the planetary boundaries discussion.”

- Food production is a central political argument for land cover changes in Brazil. This is why we assumed no land expansion.

“We used 2017 food production as the baseline for the future projections, assuming that any additional land is used after this year. This simple assumption is adopted to test the claim that more land is needed to feed the Brazilian population.”

- Regarding MapBiomas, we followed the suggestion of other reviewer and excluded the spatial explicit results from the current manuscript. We did maintain the pastureland areas calculation from MapBiomas, as explained in the Methods.

“Additionally, we obtained crop and pastureland areas for every year from 1988 to 2017 from MapBiomas’s land use and land cover maps (collection 3.0). MapBiomas is a partnership of private companies, universities and NGOs that use all available series of remotely sensed images from Landsat satellites to produce land use and land cover maps for the Brazilian territory, in the period of 1985 to 2017, with 30 metres of spatial resolution [39]. Based on the municipal data from PAM, the class “Crop or Pasture”, an error class in MapBiomas that cannot be distinguished due to spectral confusion, was allocated to “Pasturelands” or “Croplands” classes, adjusting the cropland maps to the official Brazilian cropland area. This allowed us a yearly estimate of the pasture area at the national level.”

- We use many sources of data to calculate how much nutrients the population in a given year requires and how much is produced. We presented the results in percent of the population that could be fed with the domestic production. This is found in the literature as people fed and/or nourished per hectare, as explained in the Methods. In other words, it means per capita production per hectare.

There are also a number of minor errors and style issues in the document and the manuscript would benefit from careful proof reading in advance of resubmission. For example, please see:

Line 61 - Our study provideS...

Line 153 - nutrinional - should be nutritional

Response: Thanks for the advice. We have corrected typos and checked for minor spelling errors.

---

## [Decision Letter · Decision Letter 1]

17 Mar 2021

PONE-D-20-13269R1

Is domestic agricultural production sufficient to meet national food nutrient needs in Brazil?

PLOS ONE

Dear Dr. Pompeu,

Thank you for submitting your manuscript to PLOS ONE. After careful consideration, we feel that it has merit but does not fully meet PLOS ONE’s publication criteria as it currently stands. Therefore, we invite you to submit a revised version of the manuscript that addresses the points raised during the review process.

The manuscript has greatly improved in the revision process. There remain a number of minor issues to be addressed.  Moreover, a final edit by a native-English-speaking author is also needed to catch any remaining language issues.

We look forward to receiving your revised manuscript.

Kind regards,

Gideon Kruseman, Ph.D.

Academic Editor

PLOS ONE

Journal Requirements:

Reviewers' comments:

Reviewer's Responses to Questions

**Comments to the Author**

1. If the authors have adequately addressed your comments raised in a previous round of review and you feel that this manuscript is now acceptable for publication, you may indicate that here to bypass the “Comments to the Author” section, enter your conflict of interest statement in the “Confidential to Editor” section, and submit your "Accept" recommendation.

Reviewer #1: (No Response)

Reviewer #2: (No Response)

2. Is the manuscript technically sound, and do the data support the conclusions?

Reviewer #1: Yes

Reviewer #2: Yes

3. Has the statistical analysis been performed appropriately and rigorously? 

Reviewer #1: Yes

Reviewer #2: Yes

4. Have the authors made all data underlying the findings in their manuscript fully available?

Reviewer #1: (No Response)

Reviewer #2: Yes

5. Is the manuscript presented in an intelligible fashion and written in standard English?

Reviewer #1: Yes

Reviewer #2: No

6. Review Comments to the Author

Reviewer #1: The paper is much improved with greater focus on the core question of domestic production and nutrient needs. A bit more clarification would help sharpen its main finding. Specifically, a key motivation for the paper is to counter the argument that environmental protection measures should be reduced to increase food production for the Brazilian population (line 87). The paper succeeds in doing this. But the paper then needs to note more clearly that Brazil’s importance as an agricultural exporter (lines 59, 258) suggests that export earnings are perhaps a more critical factor behind the argument for reduced environmental protection. This raises further questions about how such earnings are distributed. Line 442 notes that such earnings can improve food security for some people, but line 444 indicates these benefits may be offset by unspecified adverse social impacts, possibly including increased inequality (line 440). It would help strengthen the paper’s conclusions if the authors would come back more explicitly to what seems to be the primary argument for expanded agricultural production – namely continued growth in export earnings.

Reviewer #2: The authors addressed the major shortcomings and suggestions previously made. The article provides an interesting case against arguments that excuse land transformation in Brazil with references to domestic food security. With the changes to the manuscript this argument becomes more clear and it will appeal to a wider audience. I appreciate the authors’ answers to the previous comments, which addressed major concerns and clarified the limitations of the analysis. Only minor issues remain to be addressed in the article.

Minor comments: There are several language errors and a number of sentences are very long or wordy. I marked a number of errors and made specific suggestions below. However, these are not necessarily complete. The article could profit from additional language editing.

With regards to the terminology, I suggest being careful with the usage of the term “food demand”. A specific point is made in the comment below.

There are errors in the numbering of the citations. They are not numbered in sequence of their first citation.

Below, please find the detailed comments below:

Abstract: The authors could consider deleting the following two sentences as they are repetitive with the well explained results in the two sentences before: “We conclude that the Brazilian agriculture could deliver enough food for meet Brazilians’ nutritional needs without any additional land or changes in current diet. Nevertheless, our results show that shifting to diets with lower levels of meat consumption would substantially reduce land demand for feeding the population.” A shortened version referring back to the second sentence could be: “We conclude that no land expansion is required to produce sufficient nutrients for the Brazilian population. Food production is compatible with environmental conservation […]”

(line 42 with track changes) “Reducing the impacts of the agriculture on the environment [..]” or “We conclude that the Brazilian agriculture could deliver enough food for meet Brazilians’ nutritional needs […]” (line 57f.).

Line 42: Should be “Reducing the impact of agriculture on the environment [..]”

Lines 47-54: I suggest to separate the sentence and put the statement on GHG emissions in a second sentence.

Line 57f.: “[…] enough food to meet […]”

Line 59: “70% of global trade”, without “the”.

Line 60: For readability, I suggest: “[…]global land cropped for the international trade in agricultural commodities. This makes it the third largest exporter in the world, after the European Union and the United States and highlights its role in the global food system.”

Line 63: Split into two sentences: “biodiversity. It has the largest tropical forest area and carbon stock in the world, thereby contributing to […].”

Line 70, 74: The numbering of the citations 74 and 75 is wrong. (A few lines later the same problem repeats with number 30).

Line 83-84: The sentence is unclear. Maybe: “Despite the introduction of legislation to reduce deforestation in Brazil […]”

Line 84-86: I suggest to replace “benefited” with “encouraged”.

Line 107: For terminological reasons, I suggest to avoid the word “demand”. Demand is defined by the purchasing power and preferences of consumers, while the article refers to the required nutrients. I suggest to refer to “dietary requirements” or a similar term that specifies that the discussion centers around physiological required quantities. (Also the Article of Depenbusch and Klasen (2019, https://doi.org/10.1371/journal.pone.0223188) has a short section discussing this difference.)

Line 120: “food and non-food uses”, using “and” instead of “or”.

Line 204: The sentence appears to have a mistake and it becomes very long. I think it should be “[…] Potassium), for which we had food composition data. This allows us to access […]”

Line 255f.: Please rephrase the following sentence to make it clearer: “Domestic food calories from animal sources is always above 75% of the total animal calories produced in the period.”

Line 257: The term “all of each from dairy” is unclear. Do you mean that all animal calories used for feeds are from dairy?

Line 261-263: I would suggest to split the sentence to make it easier to read: “In 1988, calorie exportation (both plant and animal) was equivalent to 27% of all food calories. Since 2002 food calorie exports surpass domestic consumption.”

Lines 265-267: Looking at the graph, it seem that crop calories do not sum up to 100%. Could this be because losses and other usages are not plotted? If so, I recommend to just clarify this in the caption.

Line 286ff.: Suggestion for better readability: “However, as only a fraction of these calories are available as food, it suffices for less than one person per ha. Considering the production of plant and animal food products in the time span between […], on average 0.96 (±0.06) persons could be fed per ha and year.”

Line 292: Spelling should probably be “[…] at the same rate as the population”. It is interesting to see that international trade leaves just a bit less calories in the country compared to what is required. This could be misinterpreted as a situation where enough is provided for every person. As you rightly argue, due to unequal access to food this is not the case. It might be worthwhile to quote the 2020 SOFI of FAO. In Table A1.1 it specifies that about 1/5 of the population suffers from moderate or severe food insecurity.

Line 294ff.: This paragraph sums-up the provision of micro and macro nutrients but after this the new chapter starts and gets back to the historical micronutrient consumptions. I suggest to consider copying lines 301 to 316 before this paragraph. Then the following chapter focuses purely on the scenarios and not on historic values. This might also make it easier to distinguish between the produced and the domestically available micro nutrients. Currently the lines 301 ff. are a bit confusing on that point, talking about Vitamin A as the most restricting nutrient, while it still suffices for a larger population share than that quoted in lines 294ff..

Line 299: Delete empty line.

Line 311ff: I suggest to rephrase the sentence “In turn, since 2009 the produced vitamin A was sufficient for more people than the produced calories.”

Line 314: I suggest to avoid the term “demand”, maybe use “dietary requirements” here.

Line 335: Typo (“de”)

Line 338: “by comparing the curves”, without the additional “the”.

Line 341: I think “huge difference it has with a no-beef diet” should be replaced with “huge difference to a no-beef diet”.

Line 350f.: The sentence in parentheses has twice the word “but”.

Line 357f.: This seems to contradict the provision of 99% of adequacy quoted in line 294ff. for Vitamin A. Please clarify this.

Line 364f.: I suggest to just quote the inadequacy of Vitamin A and B12 here and discuss fortification as one option in the discussion.

Line 394: I suggest to shorten the sentence to improve readability “[…] by [31], who used the same sources.”

Line 397: Delete empty line.

Line 402-404: Kindly check the definition of food security in the quoted source. The FAO definition, based on the Food Insecurity Experience Scale (see page 193f. in the 2020 SOFI). By that definition, the association with “hunger” would be less clear and I would suggest to drop the reference to it in explaining food insecurity.

Line 406: I think it is more appropriate to say that the consumption of highly processed foods is a major determinant. You could for example rephrase: “As we focus on the production of food crops, we ignore later stages of the supply chain and the food environment, which have an important effect on the way food is consumed (e.g. through the consumption of highly processed foods).”

Line 408: The word “about” can be dropped.

Line 409: “natural resource preservation” without “s”

Line 418: “[…] protecting the country’s […]”

Line 418-419: The wording seems wrong “This is a complex issue that can be approached in multiple ways.”

Line 420: Wording seems wrong: “[…] can have a sizable impact on land demand for pastures, […]”

Line 421: The wording seems off and a bit long. I suggest “On the production side, pasture productivity enhancement and optimised cattle supporting rates [31, 32] can reduce the demand for pastureland.” This more careful wording would consider that productivity enhancing steps alone cannot ensure a decline in land transformation. The data of the authors shows this very nicely.

Line 431: Please clarify if this is just about international trade in agricultural commodities.

Line 432-433: I would rather say that it “reflects” the high share of raw products, as there is no strong reason why the ratio between calories and monetary value should be strongly correlated.

Line 448: I think it should be “are” not “and”.

Line 450ff.: Please rephrase the sentence and split it into two. It is hard to read and has some errors at the moment.

Line 456: Before you use “p.p.” to say per person, here it is “caput”. (the same in lines 469ff.)

I suggest to shorten the paragraph to one or two sentences, just stating that the differences are due to the exclusion of pasturelands and the inclusion of exported calories in source [36].

Line 457: “[…] based on estimates over the period 1997-2003.” Please clarify (e.g. in the previous sentence) why you refer to the year 2000.

Line 490: The word “they” is missing in “[…]although they have[…]”.

Line 500-504: The connection to incomes has been discussed before, so I suggest not to discuss it here again as shortcoming.

Line 507f.: “Finally, we did not consider the role of food imports one land demand and our diet scenarios.”

Line 514ff: I would suggest “We provide evidence that the food and nutritional needs of the Brazilian population do not necessitate an expansion of agriculture land beyond its current range.”

Line 520ff.: I suggest to delete the term “Additionally, public awareness for” to be less wordy.

Line 522: The sentence got an error. I suggest “This would allow to maintain key […]”

Lines 523-526: The sentence got a language error. Also soil conservation (though connected) has not been discussed before. Looking into sustainable agricultural practices opens a wide new area of possible policies and interventions. I suggest to either point towards them more generally or dropping the reference here.

References: Please add DOIs, where possible. Please check that the sources are quoted according to the journal standards. In particular, please correct following sources and add missing information (e.g. publisher, place published for books): 7, 13, 46, 57, 67. I recommend to use a citation software, if not done before (e.g. Zotero).

Line 738: I think the language is wrong. I suggest “Assumptions underlying the scenarios”

Table S2: The quantity for micro gram (μg with the Greek letter μ) has been replaced with ug. If the Greek letters do not work you could use mcg. Also please note for Vitamin A as unit RAE (retinol activity equivalents), behind the term for mcg, as is convention.

Table S3: I think the word “Any” should be replaced with the word “None” in this context.

7. PLOS authors have the option to publish the peer review history of their article (what does this mean?). If published, this will include your full peer review and any attached files.

Reviewer #1: No

Reviewer #2: No

---

## [Author Response · Author response to Decision Letter 1]

30 Apr 2021

Response to Reviewers

-We are grateful for the previous and current advices made by the reviewers, as well as the kind consideration of the editor. They are essential to strengthen the paper and to make our results and its implications accessible to a wider audience. We have carefully addressed all the issues raised by the two anonymous reviewers, as follows. Finally, we have gone through a careful spell check by native English speakers. 

Reviewer #1: The paper is much improved with greater focus on the core question of domestic production and nutrient needs. A bit more clarification would help sharpen its main finding. Specifically, a key motivation for the paper is to counter the argument that environmental protection measures should be reduced to increase food production for the Brazilian population (line 87). The paper succeeds in doing this. But the paper then needs to note more clearly that Brazil’s importance as an agricultural exporter (lines 59, 258) suggests that export earnings are perhaps a more critical factor behind the argument for reduced environmental protection. This raises further questions about how such earnings are distributed. Line 442 notes that such earnings can improve food security for some people, but line 444 indicates these benefits may be offset by unspecified adverse social impacts, possibly including increased inequality (line 440). It would help strengthen the paper’s conclusions if the authors would come back more explicitly to what seems to be the primary argument for expanded agricultural production – namely continued growth in export earnings.

-Response: We thank the reviewer for the encouraging commentaries and for the further suggestions. We do agree with the point raised here and recognize that it is worth mentioning it, as we did. In order to avoid a longer, we added in the paragraph that refers to the impacts of the exportation of agricultural commodities: “This reflects that most of the Brazilian agricultural exports in volume are raw products with low added value, like coffee and soybeans. These products account for nearly a fifth of the country’s GDP, with increasing trends. Therefore, the continued growth in export earnings seems to be the primary reason for expanded agricultural production rather than improving the people’s food security.[...]”

Reviewer #2: The authors addressed the major shortcomings and suggestions previously made. The article provides an interesting case against arguments that excuse land transformation in Brazil with references to domestic food security. With the changes to the manuscript this argument becomes more clear and it will appeal to a wider audience. I appreciate the authors’ answers to the previous comments, which addressed major concerns and clarified the limitations of the analysis. Only minor issues remain to be addressed in the article.

Minor comments: There are several language errors and a number of sentences are very long or wordy. I marked a number of errors and made specific suggestions below. However, these are not necessarily complete. The article could profit from additional language editing.

With regards to the terminology, I suggest being careful with the usage of the term “food demand”. A specific point is made in the comment below.

There are errors in the numbering of the citations. They are not numbered in sequence of their first citation.

Below, please find the detailed comments below:

Abstract: The authors could consider deleting the following two sentences as they are repetitive with the well explained results in the two sentences before: “We conclude that the Brazilian agriculture could deliver enough food for meet Brazilians’ nutritional needs without any additional land or changes in current diet. Nevertheless, our results show that shifting to diets with lower levels of meat consumption would substantially reduce land demand for feeding the population.” A shortened version referring back to the second sentence could be: “We conclude that no land expansion is required to produce sufficient nutrients for the Brazilian population. Food production is compatible with environmental conservation […]”

(line 42 with track changes) “Reducing the impacts of the agriculture on the environment [..]” or “We conclude that the Brazilian agriculture could deliver enough food for meet Brazilians’ nutritional needs […]” (line 57f.).

- We removed the sentences and rewrote the idea as follows: “We conclude that Brazilian agriculture could deliver enough food to meet Brazilians’ nutritional needs without further land expansion. Food production is compatible with environmental conservation in Brazil, especially if meat consumption is reduced.”

Line 42: Should be “Reducing the impact of agriculture on the environment [..]”

-Addressed in the text.

Lines 47-54: I suggest to separate the sentence and put the statement on GHG emissions in a second sentence.

-Addressed as follows: “Expansion and intensification of agriculture since the 1960s was successful in augmenting food production, but it also caused numerous adverse environmental impacts [3]. Important impacts from agriculture, though not limited to these, are habitat loss and fragmentation, with almost 40% of the ice-free land surface used for agriculture [4], growth in nitrogen fertilizer and water use of 800% and 100%, respectively [5] and emission of 23% of the total anthropogenic Greenhouse Gases (GHG). Especially, GHG emissions contribute to the current increase in global average air temperature”.

Line 57f.: “[…] enough food to meet […]”

-Addressed in the text.

Line 59: “70% of global trade”, without “the”.

-Addressed in the text.

Line 60: For readability, I suggest: “[…]global land cropped for the international trade in agricultural commodities. This makes it the third largest exporter in the world, after the European Union and the United States and highlights its role in the global food system.”

-Addressed in the text.

Line 63: Split into two sentences: “biodiversity. It has the largest tropical forest area and carbon stock in the world, thereby contributing to […].”

-Addressed in the text.

Line 70, 74: The numbering of the citations 74 and 75 is wrong. (A few lines later the same problem repeats with number 30).

- These references were inserted after the first revision and, therefore, the numbering is not continuous along the text.

Line 83-84: The sentence is unclear. Maybe: “Despite the introduction of legislation to reduce deforestation in Brazil […]”

- We think it is important to clarify that not only legislation, but government actions were important for reducing deforestation rates. Thus, we adapted the suggestion to: “Despite the introduction of legislation and governmental actions to reduce deforestation [...]”

Line 84-86: I suggest to replace “benefited” with “encouraged”.

-Addressed in the text.

Line 107: For terminological reasons, I suggest to avoid the word “demand”. Demand is defined by the purchasing power and preferences of consumers, while the article refers to the required nutrients. I suggest to refer to “dietary requirements” or a similar term that specifies that the discussion centers around physiological required quantities. (Also the Article of Depenbusch and Klasen (2019, https://doi.org/10.1371/journal.pone.0223188) has a short section discussing this difference.)

-Addressed in the whole text and referenced the suggested paper.

Line 120: “food and non-food uses”, using “and” instead of “or”.

-Addressed in the text.

Line 204: The sentence appears to have a mistake and it becomes very long. I think it should be “[…] Potassium), for which we had food composition data. This allows us to access […]”

-Addressed in the text.

Line 255f.: Please rephrase the following sentence to make it clearer: “Domestic food calories from animal sources is always above 75% of the total animal calories produced in the period.”

- We rephrased the sentence: “On the animal production side, more than 75% of the calories are for food, in the entire time series.”

Line 257: The term “all of each from dairy” is unclear. Do you mean that all animal calories used for feeds are from dairy?

- Yes. We made it clearer: “less than 5% of animal calories, which are derived from dairy products.”

Line 261-263: I would suggest to split the sentence to make it easier to read: “In 1988, calorie exportation (both plant and animal) was equivalent to 27% of all food calories. Since 2002 food calorie exports surpass domestic consumption.”

-Addressed in the text.

Lines 265-267: Looking at the graph, it seem that crop calories do not sum up to 100%. Could this be because losses and other usages are not plotted? If so, I recommend to just clarify this in the caption.

- Thanks for the advice. “Animal (red) and crop (green) calories sum 100% each, disaggregated into calories for export (dashed lines), domestic food consumption (solid lines) and animal feed (dotted lines). Wastes and other uses are not shown.”

Line 286ff.: Suggestion for better readability: “However, as only a fraction of these calories are available as food, it suffices for less than one person per ha. Considering the production of plant and animal food products in the time span between […], on average 0.96 (±0.06) persons could be fed per ha and year.”

-Addressed in the text.

Line 292: Spelling should probably be “[…] at the same rate as the population”. It is interesting to see that international trade leaves just a bit less calories in the country compared to what is required. This could be misinterpreted as a situation where enough is provided for every person. As you rightly argue, due to unequal access to food this is not the case. It might be worthwhile to quote the 2020 SOFI of FAO. In Table A1.1 it specifies that about 1/5 of the population suffers from moderate or severe food insecurity.

- We corrected the spelling. To make the readers aware of the inequality and to avoid discussions in this section (as suggested below by the reviewer regarding food fortification), we added the following sentence in red: “This provides evidence that the agricultural sector as a whole is shifting to market-oriented commodity production, while the food calorie increased at the same rate as the population. As discussed below, however, increasing food production does not reflect an even distribution of food for every person.” Also, we noted in the discussion that according to FAO, 1/5 of the population suffer from moderate or severe food insecurity: “This is illustrated by the fact that between 2014-16 and 2017-19, the number of moderately or severely food-insecure people in Brazil increased by 15%, reaching a fifth of the population [78].”

Line 294ff.: This paragraph sums-up the provision of micro and macro nutrients but after this the new chapter starts and gets back to the historical micronutrient consumptions. I suggest to consider copying lines 301 to 316 before this paragraph. Then the following chapter focuses purely on the scenarios and not on historic values. This might also make it easier to distinguish between the produced and the domestically available micro nutrients. Currently the lines 301 ff. are a bit confusing on that point, talking about Vitamin A as the most restricting nutrient, while it still suffices for a larger population share than that quoted in lines 294ff..

- We have merged the sections “Macro- and micro-nutrient production” and “Diet scenarios” and divided the paragraph to address this suggestion. We believe that this new format improves the clarity of this part. Thanks for the advice.

Line 299: Delete empty line.

-Addressed in the text.

Line 311ff: I suggest to rephrase the sentence “In turn, since 2009 the produced vitamin A was sufficient for more people than the produced calories.”

-Addressed in the text.

Line 314: I suggest to avoid the term “demand”, maybe use “dietary requirements” here.

-Addressed in the text.

Line 335: Typo (“de”)

-Corrected

Line 338: “by comparing the curves”, without the additional “the”.

-Corrected

Line 341: I think “huge difference it has with a no-beef diet” should be replaced with “huge difference to a no-beef diet”.

avoid the term “demand”, maybe use “dietary requirements” here.

-Addressed in the text.

Line 350f.: The sentence in parentheses has twice the word “but”.

-Corrected

Line 357f.: This seems to contradict the provision of 99% of adequacy quoted in line 294ff. for Vitamin A. Please clarify this.

- In line 294, we referred to 0.99 people nourished per hectare (i.e. 99 people from 100 ha). We clarified this by changing pp/ha by caput/ha.

Line 364f.: I suggest to just quote the inadequacy of Vitamin A and B12 here and discuss fortification as one option in the discussion.

- We removed the sentence “ which would require fortification and supplementation of these nutrients for an adequate vegan diet” and added in the discussion: In the BAU scenario, 220 Mha are required to produce food required by 2060, while completely shifting the diets towards a vegan diet would require only 14.5 Mha, despite limitations in vitamins A and B12 availability. Here, food fortification would play an important role for meeting the nutritional requirements of the population. Also, enhancing crop production, closing yield gaps, reducing losses and waste are strategies to deliver higher yields and increase food availability using less land [35, 65]. The redesign of the agricultural systems is critical to limit land requirements and potential expansion over natural ecosystems, regardless of dietary change.

Line 394: I suggest to shorten the sentence to improve readability “[…] by [31], who used the same sources.”

-Addressed in the text.

Line 397: Delete empty line.

-Addressed in the text.

Line 402-404: Kindly check the definition of food security in the quoted source. The FAO definition, based on the Food Insecurity Experience Scale (see page 193f. in the 2020 SOFI). By that definition, the association with “hunger” would be less clear and I would suggest to drop the reference to it in explaining food insecurity.

- We quoted the national survey that is reported to FAO, where hunger is also considered. To avoid misinterpretation, we changed the citation to FAO’s SOFI as follows: “This is illustrated by the fact that between 2014-16 and 2017-19, the number of moderately or severely food-insecure people in Brazil increased by 15%, reaching a fifth of the population [78].”

Line 406: I think it is more appropriate to say that the consumption of highly processed foods is a major determinant. You could for example rephrase: “As we focus on the production of food crops, we ignore later stages of the supply chain and the food environment, which have an important effect on the way food is consumed (e.g. through the consumption of highly processed foods).”

-Addressed in the text.

Line 408: The word “about” can be dropped.

-Addressed in the text.

Line 409: “natural resource preservation” without “s”

-Corrected

Line 418: “[…] protecting the country’s […]”

-Corrected

Line 418-419: The wording seems wrong “This is a complex issue that can be approached in multiple ways.”

-Addressed in the text.

Line 420: Wording seems wrong: “[…] can have a sizable impact on land demand for pastures, […]”

-Corrected

Line 421: The wording seems off and a bit long. I suggest “On the production side, pasture productivity enhancement and optimised cattle supporting rates [31, 32] can reduce the demand for pastureland.” This more careful wording would consider that productivity enhancing steps alone cannot ensure a decline in land transformation. The data of the authors shows this very nicely.

- Thanks for the advice. We agree and addressed this comment in the text.

Line 431: Please clarify if this is just about international trade in agricultural commodities.

- Clarified: “Regarding export of agricultural commodities, Brazilian international trade has higher shares of calories (9%–10% of global trade flows) than of monetary value (4%–5%) [10].”

Line 432-433: I would rather say that it “reflects” the high share of raw products, as there is no strong reason why the ratio between calories and monetary value should be strongly correlated.

- Thanks for the advice. We addressed in the text.

Line 448: I think it should be “are” not “and”.

-Corrected

Line 450ff.: Please rephrase the sentence and split it into two. It is hard to read and has some errors at the moment.

- We rewrote as follows: “Here, food fortification would play an important role for meeting the nutritional requirements of the population. Also, enhancing crop production, closing yield gaps, reducing losses and waste are strategies to deliver higher yields and increase food availability using less land [35, 65]. The redesign of the agricultural systems is critical to limit land requirements and potential expansion over natural ecosystems, regardless of dietary change.”

Line 456: Before you use “p.p.” to say per person, here it is “caput”. (the same in lines 469ff.)

- We now refer only to “caput”.

I suggest to shorten the paragraph to one or two sentences, just stating that the differences are due to the exclusion of pasturelands and the inclusion of exported calories in source [36].

- We addressed as follows: “To compare with [36], we found quite similar values for the year 2000 (12.2 and 3.3 caput/ha, respectively). The differences in our estimates are due to the exclusion of pasturelands and the inclusion of exports in their analysis.”

Line 457: “[…] based on estimates over the period 1997-2003.” Please clarify (e.g. in the previous sentence) why you refer to the year 2000.

- This corresponds to the data availability for the study of [36], as stated in the text: “The estimates of [36] are based on data of circa the year 2000 (1997-2003).”

Line 490: The word “they” is missing in “[…]although they have[…]”.

-Corrected

Line 500-504: The connection to incomes has been discussed before, so I suggest not to discuss it here again as shortcoming.

- Thanks for the suggestion.

Line 507f.: “Finally, we did not consider the role of food imports one land demand and our diet scenarios.”

-Addressed in the text.

Line 514ff: I would suggest “We provide evidence that the food and nutritional needs of the Brazilian population do not necessitate an expansion of agriculture land beyond its current range.”

-Addressed in the text.

Line 520ff.: I suggest to delete the term “Additionally, public awareness for” to be less wordy.

-Addressed in the text.

Line 522: The sentence got an error. I suggest “This would allow to maintain key […]”

-Addressed in the text.

Lines 523-526: The sentence got a language error. Also soil conservation (though connected) has not been discussed before. Looking into sustainable agricultural practices opens a wide new area of possible policies and interventions. I suggest to either point towards them more generally or dropping the reference here.

- We opted to drop the sentence to avoid further enlargement of the discussion section.

References: Please add DOIs, where possible. Please check that the sources are quoted according to the journal standards. In particular, please correct following sources and add missing information (e.g. publisher, place published for books): 7, 13, 46, 57, 67. I recommend to use a citation software, if not done before (e.g. Zotero).

- DOIs and links were added to the references. The mentioned references were corrected. 

Line 738: I think the language is wrong. I suggest “Assumptions underlying the scenarios”

-Addressed in the text.

Table S2: The quantity for micro gram (μg with the Greek letter μ) has been replaced with ug. If the Greek letters do not work you could use mcg. Also please note for Vitamin A as unit RAE (retinol activity equivalents), behind the term for mcg, as is convention.

-Addressed in the table.

Table S3: I think the word “Any” should be replaced with the word “None” in this context.

-Addressed in the table.

---

## [Editor Report · Decision Letter 2]

4 May 2021

Is domestic agricultural production sufficient to meet national food nutrient needs in Brazil?

PONE-D-20-13269R2

Dear Dr. Pompeu,

We’re pleased to inform you that your manuscript has been judged scientifically suitable for publication and will be formally accepted for publication once it meets all outstanding technical requirements.

Kind regards,

Gideon Kruseman, Ph.D.

Academic Editor

PLOS ONE
---

## [Editor Report · Acceptance letter]

11 May 2021

PONE-D-20-13269R2 

Is domestic agricultural production sufficient to meet national food nutrient needs in Brazil? 

Dear Dr. Pompeu:

I'm pleased to inform you that your manuscript has been deemed suitable for publication in PLOS ONE. Congratulations! Your manuscript is now with our production department. 

Kind regards, 

on behalf of

Dr. Gideon Kruseman 

Academic Editor

PLOS ONE